# UniRA: Unified Representation Alignment for Diffusion Models via Local, Structural, and Global Constraints

## Abstract

Diffusion models have achieved tremendous advancements in generative modeling generation, enabling appealing experiences in visual content generation. Yet, their conventional training objective focuses merely on predicting added noises, without any explicit consideration on the learning of intermediate features. This narrow focus might learn redundant representations that capture limited semantics and poor structural details, thus leading to suboptimal performance. To ameliorate this, this paper proposes a unified representation alignment (UniRA) paradigm that augments the diffusion objective with explicit constraints on enhancing intermediate features. Specifically, UniRA enforces three complementary forms of alignment: local semantic fidelity for discriminative patch-level features, structural consistency to preserve relational organization, and global coherence to match overall feature distributions with real data. Extensive results on the challenging ImageNet and text-to-image benchmarks show that UniRA consistently improves convergence speed and synthesis performance, gaining improved FID and precision/recall scores under the same compute budget with compared baselines. Moreover, ablative analysis demonstrate the efficacy of UniRA in reducing feature redundancy and strengthening semantic information, and improving structural organization, thereby promoting high-quality synthesis.

## 1 Introduction

Recent years have witnessed significant advancement in generative diffusion models (Ho et al., 2020; Song et al., 2021a;b; Liu et al., 2023; Lipman et al., 2022; Esser et al., 2024), setting unprecedented performance across various practical applications including image generation (Dhariwal & Nichol, 2021; Rombach et al., 2022), video synthesis Yang et al. (2024b); Blattmann et al. (2023), *etc.* The core rationale behind diffusion models is their iterative denoising process, in which the model learns to progressively reconstruct clean images from noise. In particular, the model is trained to predict the noise added to the data at different timesteps. However, despite the simplicity and stability of noise prediction, such objective overlooks intermediate feature representations in the training process, which play a critical role for generating high-quality samples. As a result, the intermediate features might contain only limited semantics, be poorly structured, and capture insufficient distributional information of the observed data, leading to suboptimal model performance.

In contrast, state-of-the-art visual foundation models, such as the DINO series (Zhang et al., 2022; Oquab et al., 2024a; Siméoni et al., 2025) and CLIP (Radford et al., 2021), have made substantial improvements by assigning auxiliary tasks on the intermediate representation rather than only predicting the final output. Specifically, the intermediate representation is encouraged to be more semantically discriminative, structurally consistent, and distributionally coherent (Chen et al., 2021; He et al., 2022). In return, the learned representations facilitate performance improvements for both recognition tasks and downstream transfer learning tasks. Such success motivates a similar question for training diffusion generative models: *are diffusion models also benefiting from explicitly enhancing the representative quality of intermediate features*?

In this paper, we seek to explore and answer this question. Our investigation is based on an intuitive premise that when the intermediate features $h_\theta(x_t, t)$ encode more information about

the clean data $x_0$, the conditional uncertainty $H(X_0 \mid h_\theta)$ decreases. That is, the intermediate features capture sufficient knowledge of the data distribution, thus reducing the modeling complexity of noise prediction, leading to less estimation variance and faster convergence. More importantly, well-structured features improve robustness since they are less sensitive to noise perturbations and generalize better across diverse inputs, facilitating diverse and high-quality synthesis. Following this philosophy, the most recent work REPA (Yu et al., 2024) aligns the intermediate features with the output of a pre-trained vision encoder (Oquab et al., 2024b). However, merely aligning patch-level features without considering the internal structural and distributional information is insufficient, leading to suboptimal performance.

To fully unlock the potential of enhancing intermediate features, we argue that a three-level alignment paradigm to regulates representation learning would be better: 1) *Local semantic alignment*, ensuring that each patch embedding contains discriminative semantics, preventing the model from learning redundant or meaningless activations; 2) *Structural consistency*, preserving structural relations between various patches, helping the model understand contours, layouts, and spatial coherence. Without structural guidance, local details may appear plausible but the global arrangement might be fragmented or distorted; 3) *Global distributional coherence*, aligning the overall representation distribution with that of real data, stabilizing training and promoting sample diversity. Without global information, models might exhibit distributional drift, leading to limited generation diversity due to mode collapse. These constraints complement each other: local constraint enhances fine-grained detail, structural constraint captures mid-level layout organization, and global constraint enforces distributional alignment. Moreover, they form a hierarchical manner for enhancing representation quality, where removing any component leads to characteristic degradation.

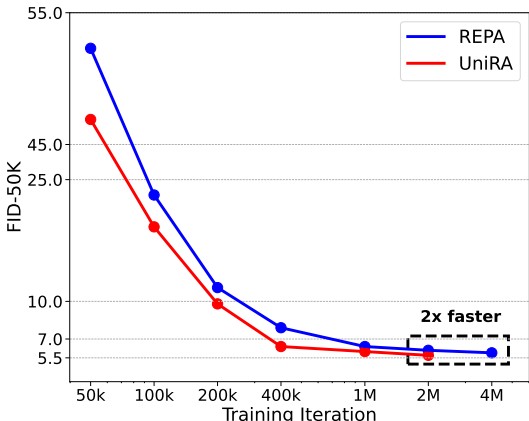

Figure 1: UniRA enhances the efficiency and effectiveness of diffusion model training through multi-level representation alignment.

Based on the above discussions, this paper proposes a unified representation alignment (UniRA) framework, for intermediate representation learning. Specifically, UniRA explicitly regulates the learning of intermediate feautres along three complementary dimensions: local semantic fidelity through patch-level alignment with pretrained encoders, structural consistency via autocorrelation alignment, and global distributional coherence through flexible distribution-matching objectives. Together, UniRA encourages the model to learn features that are simultaneously informative with fine-grained details, well-organized with clear structures, and diverse with sufficient distributional information. We conduct extensive experiments to evaluate the effectiveness of our proposed UniRA on popular generative benchmarks including ImageNet-256/512 and text-to-image datasets. The results demonstrate that UniRA consistently achieves faster convergence and superior synthesis performance compared to existing state-of-the-art baselines. Notably, UniRA achieves a new SoTA result on ImageNet $256 \times 256$ with a FID of $1.36$, as well as improved precision and recall metrics, under identical compute budgets. Additionally our analyses demonstrate why UniRA works: the intermediate features become more informative, less redundant, and more structurally organized, thus promoting more efficient denoising and higher-quality generation.

To sum up, our contributions are three folds: 1) We introduce UniRA, a training paradigm that explicitly constrains diffusion model representations to improve convergence and generative performance (as shown in Fig. 1); 2) UniRA unifies local, structural, and global alignment into a coherent objective, with each component addressing a complementary dimension of representation quality; 3) We provide extensive experimental and analytical evidence showing that UniRA reduces redundancy, strengthens semantics, and improves structural organization, ultimately bridging the gap between denoising objectives and high-quality generation.

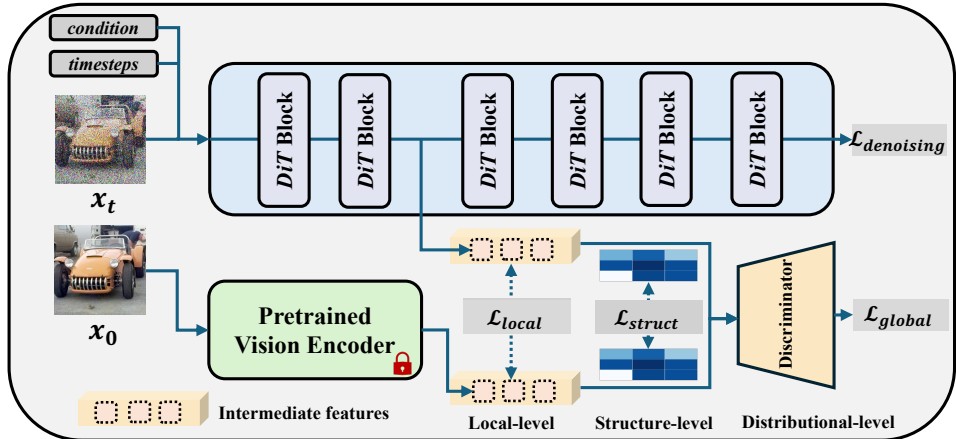

Figure 2: Overview of the UniRA framework. UniRA aligns diffusion model representations with powerful pretrained visual features through a combination of complementary alignment strategies.

## 2 RELATED WORK

**Diffusion Models.** Diffusion probabilistic models have achieved state-of-the-art results in image and video generation (Ho et al., 2020; Song et al., 2021a;b; Rombach et al., 2022; Dhariwal & Nichol, 2021; Yang et al., 2024b; Blattmann et al., 2023). Their training objective is typically formulated as predicting Gaussian noise or its velocity (Salimans & Ho, 2022), which ensures correctness of the denoising trajectory. Several studies have attempted to improve training efficiency or sample quality by modifying the objective, such as consistency models (Song et al., 2023),noise schedule optimization (Dhariwal & Nichol, 2021),and distillation-based approaches (Salimans & Ho, 2022; Song et al., 2023). However, these methods primarily regulate the output space, leaving the internal representations of the model unconstrained. Our work is complementary: we focus instead on shaping the intermediate features of the denosing model.

**Representation alignment and feature constraints.** In self-supervised representation learning, numerous methods (Chen et al., 2020; Zhang et al., 2022; Oquab et al., 2024a) demonstrate that enforcing invariances and structural consistency leads to discriminative and transferable features. These insights have motivated works that incorporate external representation signals into generative training (Pernias et al., 2023; Li et al., 2025). For example, REPA (Yu et al., 2024) aligns patch embeddings of diffusion denoisers with pretrained encoder features to improve semantic fidelity. While effective, such methods focus only on local alignment. Our work generalizes this idea into a unified framework that integrates local, structural, and global constraints, offering complementary benefits for representation quality and generation fidelity.

**Distributional alignment in generative modeling.** Generative adversarial networks (Goodfellow et al., 2020; Johnson et al., 2016) align generated and real distributions through adversarial objectives but often suffer from instability and mode collapse (Arjovsky et al., 2017). Alternative distribution matching losses, such as maximum mean discrepancy (Gretton et al., 2012) and sliced Wasserstein distance (Rabin et al., 2011), have also been applied in generative modeling. To stabilize training, several works explore adversarial regularization within diffusion frameworks (Yang et al., 2024a), or combine diffusion processes with GAN training (Wang et al., 2022). In this work, we adopt a lightweight adversarial module for global distributional alignment, as it provides a practical balance between effectiveness and efficiency in large-scale diffusion training.

## 3 THE PROPOSED UNIRA

### 3.1 OVERALL FRAMEWORK

Fig. 2 presents the overall framework of our proposed UniRA. Specifically, UniRA builds upon a standard diffusion model with denoising network $f_\theta$. To enable representation alignment, we incorporate three auxiliary components: 1) A frozen pretrained encoder (e.g., DINOv2) that produces

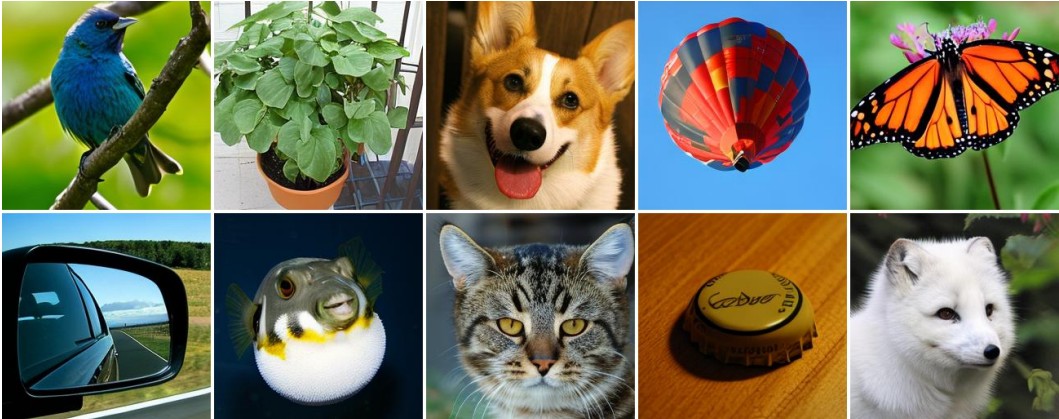

Figure 3: Generated samples from the SiT-XL/2+UniRA model on ImageNet $256 \times 256$ using classifier-free guidance with $w = 4.0$. More visual results are provided in the supplementary.

reference features from clean inputs. These features serve as semantic and structural anchors. Importantly, the encoder is never updated during training. 2) A lightweight projection head $\phi$, applied only to denoiser features, maps them into a common alignment space. Reference features are used as-is, without transformation. 3) A compact discriminator, introduced only for global alignment, distinguishes pooled denoiser features from pooled reference features. The discriminator is small and updated intermittently to keep overhead minimal.

Together with the standard denoising loss, UniRA introduces three complementary objectives: local semantic alignment, structural consistency, and global distributional coherence, jointly enhancing the quality of intermediate representations. This yields a unified framework that reduces redundancy, enriches semantics, and improves structural organization, ultimately leading to higher-fidelity generations. Further training object details of the denoising network are provided in Appendix E.

## 3.2 LOCAL SEMANTIC ALIGNMENT

Denoiser features at the patch level often collapse into redundant activations, limiting their ability to encode meaningful semantics. To encourage semantic richness, we align patch embeddings from the denoiser with reference features extracted from the frozen encoder.

Let the feature map at layer $l$ produce $N$ patch embeddings $H = [h_1, \ldots, h_N]$. The encoder provides corresponding reference features $Z = [z_1, \ldots, z_N]$. A projection head $\phi$ maps denoiser features to $\tilde{h}_i = \phi(h_i)$, while reference features are used directly as $\tilde{z}_i = z_i$.

The local alignment loss is defined as a cosine similarity objective:

$$\mathcal{L}_{\text{local}} = \mathbb{E}_{x_0,t}\left[ \frac{1}{N} \sum_{i=1}^{N} \left( 1 - \frac{\langle \tilde{h}_i, \tilde{z}_i \rangle}{\|\tilde{h}_i\|\|\tilde{z}_i\|} \right) \right]. \tag{1}$$

This loss encourages patch-level representations to capture discriminative semantics consistent with the reference encoder.

## 3.3 STRUCTURAL CONSISTENCY

While local alignment improves semantic fidelity of individual patches, it does not constrain the relationships among patches, which are crucial for representing object structure and spatial organization. To address this, we align relational patterns using similarity matrices. Given projected features $\tilde{H} \in \mathbb{R}^{N \times p}$, we compute a normalized self-similarity matrix:

$$A(\tilde{H}) = \frac{\tilde{H}\tilde{H}^\top}{\|\tilde{H}\tilde{H}^\top\|_F}. \tag{2}$$

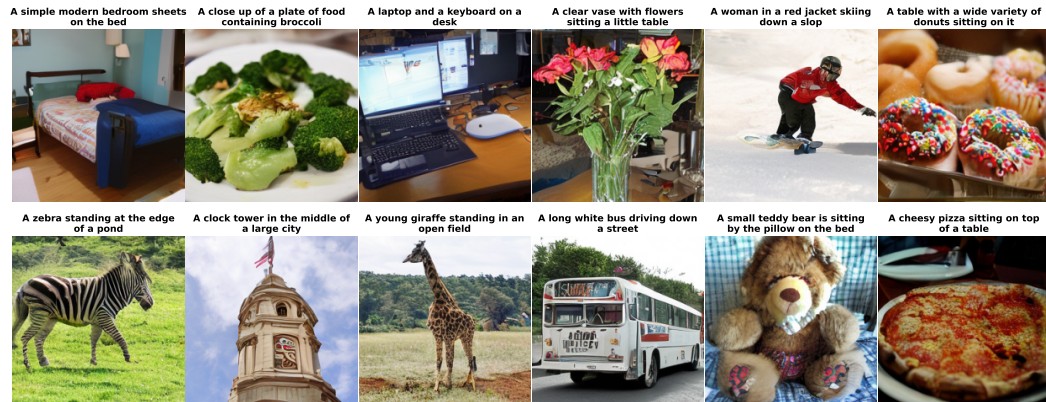

Figure 4: Qualitative comparison on text-to-image generation (MS-COCO). We use classifier-free guidance with $w = 4.0$.

The same computation yields $A(Z)$ from reference features. The structural alignment loss minimizes the Frobenius distance between the two:

$$\mathcal{L}_{\text{struct}} = \mathbb{E}_{x_0, t}\left[\|A(\tilde{H}) - A(Z)\|_F^2\right]. \tag{3}$$

By enforcing relational consistency, this objective ensures that denoiser features not only encode semantics per patch but also preserve coherent spatial and structural organization.

### 3.4 GLOBAL DISTRIBUTIONAL COHERENCE

Even with local and structural constraints, feature distributions may still exhibit mismatch, leading to mode imbalance or unnatural style shifts. To complement patch-level and relational alignment, UniRA optionally performs feature-level distributional alignment. We realize this via a compact discriminator that operates on pooled denoiser features and frozen encoder features.

Specifically, We pool patch features into global vectors: $\hat{H} = \text{POOL}(\tilde{H})$, $\hat{Z} = \text{POOL}(Z)$. A discriminator $D_\phi$ is trained to distinguish pooled reference features from denoiser features:

$$\min_\theta \max_\phi \ \mathbb{E}_{x_0}[\log D_\phi(\hat{Z})] + \mathbb{E}_{x_0, t}[\log(1 - D_\phi(\hat{H}))]. \tag{4}$$

The generator-side loss is:

$$\mathcal{L}_{\text{global}} = \mathbb{E}_{x_0, t}\left[-\log D_\phi(\hat{H})\right]. \tag{5}$$

The global objective provides an additional, complementary signal that refines the model's representation distribution and yields measurable improvements in the reported generation metrics when combined with the local and structural losses. Because the primary gains of UniRA arise from the local and structural components, we treat the global discriminator as an optional refinement that can be enabled when slight additional improvements are desired.

### 3.5 JOINT TRAINING AND OPTIMIZATION

Our final training objective combines all alignment losses and denoising loss:

$$\mathcal{L}(\theta) = \mathcal{L}_{\text{denoise}} + \lambda \mathcal{L}_{\text{local}} + \beta \mathcal{L}_{\text{struct}} + \gamma \mathcal{L}_{\text{global}}, \tag{6}$$

where $\lambda$, $\beta$, and $\gamma$ control the relative strength of the three constraints. UniRA thereby unifies local semantics, structural relations, and global distributions toward high-fidelity generation.

## 4 EXPERIMENTS

In this part, we conduct extensive experiments to validate the effectiveness of UniRA. Our experiments are designed to examine whether the proposed representation alignment improves the generative performance of diffusion models, how each of the three constraints contributes to the overall

Table 1: Quantitative comparison of UniRA, REPA, and other diffusion models on ImageNet 256×256. ↓ and ↑ indicate whether lower or higher values are preferable, respectively.

(a) FID comparisons with DiTs, SiTs, and REPA without classifier-free guidance (CFG). Iter. denotes the training iteration.

| Model | #Params | Iter. | FID↓ |
|---|---|---|---|
| DiT-L/2 | 458M | 400K | 23.3 |
| + REPA | 458M | 400K | 15.6 |
| **+ UniRA** | **458M** | **400K** | **13.5** |
| DiT-XL/2 | 675M | 400K | 19.5 |
| + REPA | 675M | 400K | 12.3 |
| **+ UniRA** | **675M** | **400K** | **10.3** |
| SiT-B/2 | 130M | 400K | 33.0 |
| + REPA | 130M | 400K | 24.4 |
| **+ UniRA** | **130M** | **400K** | **22.1** |
| SiT-L/2 | 458M | 400K | 18.8 |
| + REPA | 458M | 400K | 10.0 |
| **+ UniRA** | **458M** | **400K** | **8.5** |
| SiT-XL/2 | 675M | 7M | 8.3 |
| + REPA | 675M | 400K | 7.9 |
| **+ UniRA** | **675M** | **400K** | **6.4** |
| + REPA | 675M | 2M | 6.1 |
| **+ UniRA** | **675M** | **1M** | **6.0** |
| + REPA | 675M | 4M | 5.9 |
| **+ UniRA** | **675M** | **2M** | **5.7** |

(b) Evaluate on ImageNet 256×256 with classifier-free guidance (CFG). Results that include additional CFG scheduling are marked with an asterisk (*), where the guidance interval from is applied for REPA and UniRA.

| Model | Epochs | FID↓ | sFID↓ | IS↑ | Prec.↑ | Rec.↑ |
|---|---|---|---|---|---|---|
| ADM-U (Dhariwal & Nichol, 2021) | 400 | 3.94 | 6.14 | 186.7 | 0.82 | 0.52 |
| VDM++ (Kingma & Gao, 2023) | 560 | 2.40 | - | 225.3 | - | - |
| Simple diffusion (Hoogeboom et al., 2023) | 800 | 2.77 | - | 211.8 | - | - |
| CDM (Ho et al., 2022) | 2160 | 4.88 | - | 158.7 | - | - |
| LDM-4 (Rombach et al., 2022) | 200 | 3.6 | - | 247.7 | 0.87 | 0.48 |
| U-ViT-H/2 (Bao et al., 2023) | 240 | 2.29 | 5.68 | 263.9 | 0.82 | 0.57 |
| DiffiT* (Hatamizadeh et al., 2024) | - | 1.73 | - | 276.5 | 0.80 | 0.62 |
| MDTv2-XL/2* (Gao et al., 2023) | 1080 | 1.58 | 4.52 | 314.7 | 0.79 | 0.65 |
| MaskDiT (Zheng et al., 2023) | 1600 | 2.28 | 5.67 | 276.6 | 0.80 | 0.61 |
| SD-DiT (Zhu et al., 2024) | 480 | 3.23 | - | - | - | - |
| DiT-XL/2 | 1400 | 2.27 | 4.60 | 278.2 | 0.83 | 0.57 |
| SiT-XL/2 | 1400 | 2.06 | 4.50 | 270.3 | 0.82 | 0.59 |
| + REPA | 200 | 1.96 | 4.49 | 264.0 | 0.82 | 0.60 |
| **+ UniRA** | **200** | **1.82** | 4.51 | **279.8** | **0.83** | **0.60** |
| + REPA | 800 | 1.80 | 4.50 | 284.0 | 0.81 | 0.61 |
| **+ UniRA** | **400** | **1.75** | **4.48** | **288.9** | **0.82** | **0.61** |
| + REPA* | 800 | 1.42 | 4.70 | 305.7 | 0.80 | **0.65** |
| **+ UniRA*** | **400** | **1.36** | **4.63** | **316.7** | **0.81** | 0.63 |

Table 2: Evaluated on ImageNet 512×512 using classifier-free guidance with $\omega = 1.35$.

| Model | Epochs | FID↓ | sFID↓ | IS↑ | Pre.↑ | Rec.↑ |
|---|---|---|---|---|---|---|
| VDM++ | - | 2.65 | - | 278.1 | - | - |
| ADM-G,ADM-U | 400 | 2.85 | 5.86 | 2217 | 0.84 | 0.53 |
| Simple diffusion (U-Net) | 800 | 4.28 | - | 171.0 | - | - |
| Simple diffusion (U-ViT, L) | 800 | 4.53 | - | 205.3 | - | - |
| MaskDiT | 800 | 2.50 | 5.10 | 256.3 | 0.83 | 0.56 |
| DiT-XL/2 | 600 | 3.04 | 5.02 | 240.8 | 0.84 | 0.54 |
| SiT-XL/2 | 600 | 2.62 | 4.18 | 252.2 | 0.84 | 0.54 |
| + REPA | 100 | 2.32 | 4.16 | 255.7 | 0.84 | 0.56 |
| **+ UniRA** | **100** | **2.14** | **4.11** | **266.8** | **0.84** | **0.57** |
| + REPA | 200 | 2.08 | 4.19 | 274.6 | 0.83 | 0.58 |
| **+ UniRA** | **200** | **1.93** | 4.15 | **287.7** | **0.84** | **0.58** |

Table 3: Evaluated on T2I generation with CFG($\omega = 2.0$), following REPA

| Method | Type | FID↓ |
|---|---|---|
| DM-GAN (Zhu et al., 2019) | GAN | 32.64 |
| VQ-Diffusion (Gu et al., 2022) | Discrete Diffusion | 19.75 |
| DF-GAN (Tao et al., 2022) | GAN | 19.32 |
| XMC-GAN (Zhang et al., 2021) | GAN | 9.33 |
| Frido (Fan et al., 2023) | Diffusion | 8.97 |
| U-Net (Bao et al., 2023) | Diffusion | 7.32 |
| U-ViT-S/2(Deep) (Bao et al., 2023) | Diffusion | 5.48 |
| MMDiT(ODE;NFE=50) | Diffusion | 6.05 |
| MMDiT+REPA(ODE;NFE=50) | Diffusion | 4.73 |
| **MMDiT+UniRA(ODE;NFE=50)** | Diffusion | **4.11** |
| MMDiT(ODE;NFE=250) | Diffusion | 5.30 |
| MMDiT+REPA(ODE;NFE=250) | Diffusion | 4.14 |
| **MMDiT+UniRA(ODE;NFE=250)** | Diffusion | **3.67** |

gain, and what representational changes underlie these improvements. To this end, we evaluate UniRA on multiple image generation benchmarks, comparing against strong diffusion baselines under the same training settings.

Specifically, Section 4.1 details the experimental setup. Section 4.2 report quantitative results on standard metrics, including FID, IS, and precision–recall, to establish the overall benefit of UniRA. Section 4.3 then presents ablation studies by selectively removing each alignment component and measuring the resulting performance differences, which highlight the complementary roles of local, structural, and global objectives. Section 4.4 perform detailed representation analyses to probe the internal effects of UniRA. These analyses reveal how UniRA reduces feature redundancy, enhances semantic information, and improves structural organization, thereby bridging the gap between denoising accuracy and perceptual fidelity.

## 4.1 SETUP

**Implementation details.** Our experimental setup closely follows REPA (Yu et al., 2024), which builds upon DiT (Peebles & Xie, 2023) and SiT (Ma et al., 2024), unless stated otherwise. We use ImageNet (Deng et al., 2009), preprocessing images to 256×256 and 512×512 resolution following the data protocols of ADM (Dhariwal & Nichol, 2021). The images are mapped into a compressed latent representation, $\mathbf{z} \in \mathbb{R}^{32 \times 32 \times 4}$ ($\mathbf{z} \in \mathbb{R}^{64 \times 64 \times 4}$ for 512 resolution), using the Stable Diffusion VAE (Rombach et al., 2022). For model configurations, we adopt the B/2, L/2, and XL/2 architectures from DiT and SiT, using a patch size of 2. To ensure a fair comparison with DiT, SiT, and REPA, we maintain a fixed batch size of 256 during training. For adversarial alignment, we use the first two blocks of a pretrained ResNet-18 (He et al., 2016) as the discriminator to distinguish between representations from the pretrained vision encoder and the diffusion model. To match the discriminator's input dimensions, we apply a randomly initialized convolutional layer to adjust rep-

Table 4: Ablation study for alignment strategies

| + Local | + Struc | + Global | FID↓ | IS↑ |
|---|---|---|---|---|
| ✓ | | | 7.9 | 122.6 |
| ✓ | ✓ | | 7.3 | 131.9 |
| ✓ | | ✓ | 7.9 | 121.5 |
| | ✓ | ✓ | 8.4 | 118.1 |
| ✓ | ✓ | ✓ | 6.4 | 138.8 |

Table 5: Ablation study for $\beta$ and $\gamma$

| $\beta$ | $\gamma$ | FID ↓ | IS↑ |
|---|---|---|---|
| 0.1 | 0.01 | 7.9 | 123.1 |
| 0.1 | 0.05 | 7.7 | 125.6 |
| 0.5 | 0.01 | 7.2 | 130.4 |
| 0.5 | 0.05 | 6.4 | 138.8 |
| 0.5 | 0.1 | 7.3 | 127.7 |
| 1.0 | 0.1 | 7.7 | 126.4 |

resentation dimensions before feeding them into the network. For inference, we adopt the SDE Euler-Maruyama sampler and set the default number of function evaluations (NFE) to 250. More details including hyperparameters and computational resources are provided in Appendix B.

**Evaluation Metrics.** We evaluate our method using 50,000 samples and report Frechet Inception Distance (FID) (Heusel et al., 2017), sFID (Nash et al., 2021), Inception Score (IS) (Salimans et al., 2016), Precision (Pre.), and Recall (Rec.) (Kynkäänniemi et al., 2019).

## 4.2 MAIN RESULTS

We present a comprehensive evaluation of various DiT and SiT models trained with UniRA, analyzing their performance across multiple benchmarks. Additionally, we perform a system-level comparison against recent state-of-the-art diffusion models and diffusion transformers trained with REPA, demonstrating the effectiveness of our proposed alignment strategy. All models are aligned with DINOv2-B representations using $\lambda = 0.5$, $\beta = 0.5$, and $\gamma = 0.05$. For the base model, we use the 4th-layer hidden states, while for the large and xlarge models, we use the 8th-layer hidden states. We conduct a detailed comparison under two settings: without classifier-free guidance (w/o CFG) and with classifier-free guidance (CFG). For each setting, we provide both quantitative evaluations and qualitative assessments of the generated results.

**W/o CFG.** As shown in Fig. 1, under the SiT-XL/2 configuration, UniRA consistently achieves lower FID scores than REPA across training iterations. Notably, at 2M iterations, UniRA attains an FID of 5.7, surpassing REPA's best performance at 4M iterations (FID = 6.1). As summarized in Tab. 1a, UniRA outperforms REPA across all model variants, demonstrating its effectiveness in improving generation quality across diffusion transformers of varying scales and architectures.

**With CFG.** We evaluate SiT-XL/2 under classifier-free guidance (CFG) and compare it quantitatively with recent state-of-the-art diffusion models. Using a fixed guidance scale $w = 1.35$ (as in REPA) without extensive tuning, UniRA achieves comparable performance to REPA at 200 epochs while requiring $4\times$ fewer epochs, and surpasses the original SiT-XL/2 with $7\times$ fewer. At 400 epochs, SiT-XL/2 trained with UniRA attains an FID of 1.75, outperforming REPA with half the training epochs; further CFG tuning reduces the FID to 1.36. Tab. 2 shows UniRA scales effectively with image resolution: at 512, it matches REPA with half the epochs and eventually reaches a state-of-the-art FID of 1.93. Fig. 3 presents qualitative results from SiT-XL/2 trained with UniRA, demonstrating its improved synthesis quality. More examples are provided in Appendix H.

**Text-to-Image Generation.** We further evaluate the effectiveness of UniRA in the text-to-image (T2I) generation setting. Unless otherwise noted, we adopt the same experimental protocol as REPA (Yu et al., 2024): models are trained from scratch on the MS-COCO training split (Lin et al., 2014), with evaluation conducted on the validation split. We employ MMDiT (Esser et al., 2024), a simplified DiT variant that integrates attention over both image patches and text embeddings. The MMDiT models are trained for 150K iterations with a batch size of 256, using a hidden dimension of 768 and a depth of 24 layers. Text prompts are encoded using a CLIP (Radford et al., 2021) text encoder. As shown in Tab. 3, UniRA leads to substantial performance gains in T2I generation, underscoring the effectiveness of aligning visual representations even in the presence of strong textual guidance. The qualitative results from MMDiT trained with UniRA are provided in Fig. 4.

## 4.3 ABLATION STUDY

**Effect of Alignment Strategies.** We first examine the contribution of each alignment component in UniRA by systematically evaluating different combinations of local semantic alignment (Local), structural consistency (Struct), and global distributional coherence (Global). Models are trained for

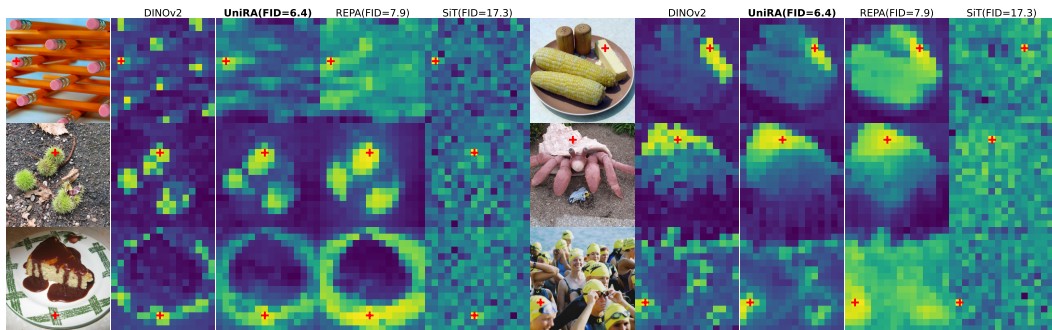

Figure 5: Structural correlation heatmaps of intermediate representations. Heatmaps illustrating the correlation of a selected patch (marked with a red cross) within the intermediate representations of DINOv2, UniRA, REPA and SiT.

400K iterations with coefficient $\lambda = 0.5$, $\beta = 0.5$, $\gamma = 0.05$ whenever the corresponding term is activated. Results are summarized in Tab. 4.

Using local alignment alone provides a strong baseline, since directly matching patch-level features with the reference encoder significantly improves semantic fidelity. Adding structural alignment on top of the local term further enhances representation consistency and perceptual quality. Interestingly, applying structural and global alignment without local alignment produces smaller gains. This is consistent with our hypothesis that local semantics provide the foundation upon which relational and distributional regularization can be effective; without meaningful local anchors, enforcing structure or global coherence becomes less reliable. The full UniRA model, which integrates all three components, achieves the best overall performance, highlighting the complementary nature of the proposed alignment objectives. In practice, enabling the global term can be used as a refinement step to squeeze additional gains from representation-level distributional matching, but the core benefits of UniRA are achieved already with the local and structural objectives.

**Effect of Coefficients.** We next analyze the sensitivity to the structural and global weights $\beta$ and $\gamma$. In these experiments, we fix the local weight to $\lambda = 0.5$ and vary $\beta$ and $\gamma$ (Tab. 5). This design choice reflects our empirical observation that local alignment is the most stable and essential component; keeping its contribution fixed allows us to more clearly isolate the role of the structural and global terms. We observe consistent improvements as $\beta$ and $\gamma$ increase, reaching optimal performance around $\beta = 0.5$ and $\gamma = 0.05$. While generation quality can be sensitive to the choice of $\lambda$, $\beta$, $\gamma$, we find that effective ranges are reasonably broad, and satisfactory performance can be achieved without exhaustive tuning. Moreover, the three terms exhibit complementary roles: local alignment contributes semantic fidelity, structural alignment enforces spatial organization, and global alignment improves distributional coverage. This modularity facilitates adaptation to different datasets or domains, as tuning can often be limited to adjusting one or two coefficients.

**Effect of Pretrained Encoder.** We investigate the effect of pretrained encoders on UniRA through experiments on ImageNet-256, examining how encoder type, model size, and feature extraction depth influence performance. Detailed results are presented in Appendix C.

## 4.4 REPRESENTATION ANALYSES

To further understand why UniRA improves generation quality, we analyze the internal representations of the denoising network under different training strategies. While quantitative metrics such as FID and IS establish the overall benefits, representation analyses provide insight into the mechanism: UniRA improves structural organization, enriches semantic features, and reduces redundancy by aligning denoiser features with pretrained encoders at multiple levels. Detailed experimental configurations for each analysis are provided in Appendix D.

**Structural organization of features.** We qualitatively analyze structural relations in intermediate features by selecting a reference patch from the denoiser's mid-level layer and computing its similarity with all other patches. Fig. 5 shows six representative cases comparing DINOv2, UniRA, REPA, and standard diffusion training (SiT). The pretrained DINOv2 encoder serves as a reference,

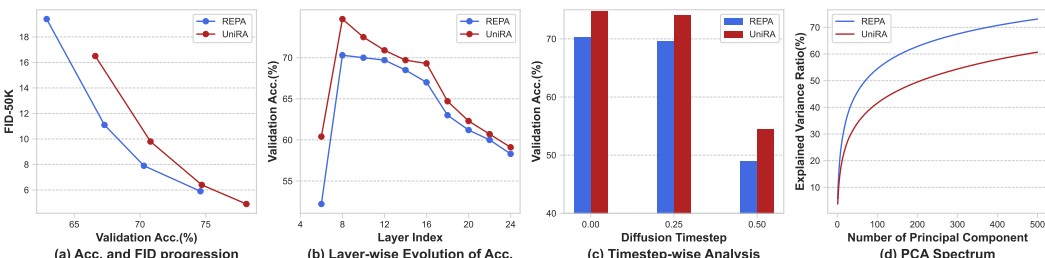

Figure 6: Structural correlation heatmaps of intermediate representations. Heatmaps illustrating the correlation of a selected patch (marked with a red cross) within the intermediate representations of DINOv2, UniRA, REPA and SiT.

exhibiting sharp locality and meaningful long-range correlations. UniRA closely recovers these patterns, while REPA and SiT yield blurrier, noisier maps. The degradation from left to right highlights how structural coherence weakens without alignment and is effectively restored by UniRA.

**Semantic predictability and generation quality.** To directly link representation quality to downstream generation, we evaluate linear probes on frozen intermediate features and compare their classification accuracy with the model's FID. As shown in Fig. 6(a), higher probe accuracy strongly correlates with lower FID. UniRA consistently achieves both higher accuracy and lower FID than REPA, confirming that semantically richer representations translate into better perceptual fidelity.

**Layer-wise analysis.** Fig. 6(b) shows the progression of probe accuracy across network layers. Semantic predictability follows a typical pattern: accuracy gradually increases from shallow to mid layers, then saturates and slightly decreases toward the output. UniRA achieves consistently higher accuracy at every layer. This improvement reflects the complementary nature of UniRA's objectives. By jointly enforcing local, structural, and global alignment, UniRA strengthens semantic information across all layers, leading to consistently higher probe accuracy than REPA. While the precise contribution of each objective to specific depths may vary, the overall effect is clear: semantic representations are preserved and enhanced more effectively throughout the hierarchy.

**Timestep robustness.** Another important dimension is the temporal trajectory of denoising. Fig. 6(c) shows probe accuracy when features are extracted at different timesteps. UniRA maintains much stronger semantic predictability, suggesting that alignment prevents the model from losing semantic grounding even in heavily corrupted states. This robustness ensures that denoising remains guided by meaningful content throughout the process.

**Redundancy in representations.** To quantify redundancy in learned features, we extract 10000 intermediate-layer representations from the ImageNet validation set and apply PCA analysis. Fig. 6(d) plots the cumulative explained variance ratio of the top-n principal components for both REPA and UniRA. The REPA curve rises steeply, with a large fraction of variance captured by only a few leading components, indicating strong redundancy and reduced effective capacity. By contrast, the UniRA curve grows more gradually and remains consistently lower, showing that variance is distributed across a broader set of components. This demonstrates that UniRA produces more expressive and less redundant representations, supporting the claim that its alignment objectives prevent collapse into narrow subspaces and enable richer feature organization for generation.

## 5 CONCLUSIONS

In this work, we presented UniRA, a unified representation alignment framework for diffusion models. By jointly enforcing local semantic alignment, structural consistency, and global distributional coherence, UniRA provides complementary constraints that improve the internal representations of the denoising network. Extensive experiments show that this approach consistently enhances generation quality across standard benchmarks, and detailed analyses reveal how UniRA reduces redundancy, enriches semantic features, and improves feature organization. We believe these findings highlight the promise of representation-level constraints as a general principle for advancing diffusion-based generative models. Looking forward, extending UniRA to more complex domains such as high-resolution synthesis, video, and multimodal generation offers exciting avenues for future exploration.

ETHICS STATEMENT

This work focuses on improving the training of diffusion-based generative models by introducing representation alignment strategies. Our contributions are primarily methodological, aiming to enhance sample quality and efficiency. As with other generative modeling research, potential ethical concerns include misuse for generating misleading or harmful visual content. While our experiments are limited to standard benchmark datasets such as ImageNet, we acknowledge that real-world applications require safeguards against misuse. We believe our research is best applied to scientific and creative domains where higher-quality generation can support downstream innovation, and we encourage responsible deployment aligned with ethical guidelines in generative AI.

REPRODUCIBILITY STATEMENT

We have made every effort to ensure reproducibility. The paper provides detailed descriptions of the UniRA framework, including objectives, training setup, and hyperparameters. Experimental configurations, such as dataset preprocessing, model variants, and optimization details, are specified in Section 4 and Appendix B. Representation analyses are described step by step in Appendix D. All baselines follow standard open-source implementations to ensure comparability. To further support reproducibility, we will make all of our code, pretrained models publicly available.

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

Table A1: Hyperparameter setup of our experiments.

| | Figure 2 | Table 1a (SiT-B) | Table 1a (SiT-L) | Table 1a (SiT-XL) | Table 1b |
|---|---|---|---|---|---|
| **Architecture** | | | | | |
| Input dim. | $32 \times 32 \times 4$ | $32 \times 32 \times 4$ | $32 \times 32 \times 4$ | $32 \times 32 \times 4$ | $32 \times 32 \times 4$ |
| Layers | 28 | 12 | 24 | 28 | 28 |
| Hidden dim. | 1152 | 768 | 1024 | 1152 | 1152 |
| Heads | 16 | 12 | 16 | 16 | 16 |
| **UniRA** | | | | | |
| $\lambda$ | 0.5 | 0.5 | 0.5 | 0.5 | 0.5 |
| $\beta$ | 0.5 | 0.5 | 0.5 | 0.5 | 0.5 |
| $\gamma$ | 0.05 | 0.05 | 0.05 | 0.05 | 0.05 |
| Alignment depth | 8 | 4 | 8 | 8 | 8 |
| $sim(\cdot, \cdot)$ | cos. sim. | cos. sim. | cos. sim. | cos. sim. | cos. sim. |
| Encoder $f(x)$ | DINOv2-B | DINOv2-B | DINOv2-B | DINOv2-B | DINOv2-B |
| **Optimization** | | | | | |
| Training iteration | 400K | 400K | 400K | 2M | 2M |
| Batch size | 256 | 256 | 256 | 256 | 256 |
| Optimizer | AdamW | AdamW | AdamW | AdamW | AdamW |
| lr | 0.0001 | 0.0001 | 0.0001 | 0.0001 | 0.0001 |
| $(\beta_1, \beta_2)$ | (0.9, 0.999) | (0.9, 0.999) | (0.9, 0.999) | (0.9, 0.999) | (0.9, 0.999) |
| **Interpolants** | | | | | |
| $\alpha_t$ | $1 - t$ | $1 - t$ | $1 - t$ | $1 - t$ | $1 - t$ |
| $\sigma_t$ | $t$ | $t$ | $t$ | $t$ | $t$ |
| $w_t$ | $\sigma_t$ | $\sigma_t$ | $\sigma_t$ | $\sigma_t$ | $\sigma_t$ |
| Training objective | v-prediction | v-prediction | v-prediction | v-prediction | v-prediction |
| Sampler | Euler-Maruyama | Euler-Maruyama | Euler-Maruyama | Euler-Maruyama | Euler-Maruyama |
| Sampling steps | 250 | 250 | 250 | 250 | 250 |
| Guidance | - | - | - | - | 1.35 |

# A USE OF LARGE LANGUAGE MODELS

In preparing this manuscript, we used large language models (LLMs) solely to assist with language polishing and minor grammatical refinements. All research ideas, methodological designs, experiments, and substantive writing were conceived, conducted, and authored by the paper's contributors. The LLM was not involved in data analysis, experimental design, or the generation of technical content.

# B HYPERPARAMETERS AND MORE IMPLEMENTATION DETAILS

**Further implementation details.** To ensure a fair comparison, our experimental setup is nearly identical to REPA. Specifically, we adopt the same architecture as DiT and SiT throughout all experiments. We use AdamW as the optimizer with a learning rate of 1e-4, $(\beta_1, \beta_2) = (0.9, 0.999)$, and no weight decay. To accelerate training, we employ mixed-precision (fp16) training with gradient clipping and precompute compressed latent vectors from raw pixels using Stable Diffusion VAE, which are then used throughout training. For the MLP used in projection, we adopt a three-layer MLP with SiLU activation. A detailed hyperparameter configuration is provided in Table A1.

**Discriminator details.** For adversarial alignment training, we employ a lightweight discriminator to minimize computational overhead. Specifically, we use a pretrained ResNet-18, removing its final two resdual blocks to reduce complexity. Additionally, we modify the first convolutional layer to a $1 \times 1$ convolution, allowing it to transform transformer-based representations into a suitable input dimension for the discriminator. During training, the discriminator distinguishes between features extracted by the diffusion model (negative samples) and features from the pretrained vision encoder (positive samples). Since the discriminator is not trained from scratch, we adopt an asymmetric training strategy—for every five updates to the diffusion model, the discriminator is updated only

Table A2: Ablation study of pretrained encoders on ImageNet-256. All models are SiT-L/2 trained for 400K iterations. All metrics are measured with the SDE Euler-Maruyama sampler with NFE=250 and without classifier-free guidance. We fix $\lambda = 0.5, \beta = 0.5, \gamma = 0.05$ here. $\downarrow$ and $\uparrow$ indicate whether lower or higher values are better, respectively.

| Target Repr. | Depth | FID↓ | sFID↓ | IS↑ | Pre.↑ | Rec.↑ |
|---|---|---|---|---|---|---|
| SiT-L/2 | 8 | 18.8 | 5.29 | 72.9 | 0.64 | 0.64 |
| MAE-L+REPA | 8 | 12.5 | 4.89 | 90.7 | 0.68 | 0.63 |
| **MAE-L+UniRA** | 8 | 11.5 | 4.97 | 102.3 | 0.69 | 0.64 |
| MoCov3-L+REPA | 8 | 11.9 | 5.06 | 92.2 | 0.68 | 0.64 |
| **MoCov3-L+UniRA** | 8 | 11.1 | 5.02 | 98.8 | 0.69 | 0.64 |
| CLIP-L+REPA | 8 | 11.0 | 5.25 | 107.0 | 0.69 | 0.64 |
| **CLIP-L+UniRA** | 8 | 9.8 | 5.15 | 110.8 | 0.70 | 0.64 |
| DINOv2-B+REPA | 8 | 9.7 | 5.13 | 107.5 | 0.69 | 0.64 |
| **DINOv2-B+UniRA** | 8 | 8.5 | 5.24 | 119.3 | 0.69 | 0.66 |
| DINOv2-L+REPA | 8 | 10.0 | 5.09 | 106.6 | 0.68 | 0.65 |
| **DINOv2-L+UniRA** | 8 | 8.4 | 5.18 | 121.6 | 0.69 | 0.66 |
| DINOv2-g+REPA | 8 | 9.8 | 5.22 | 108.9 | 0.69 | 0.64 |
| **DINOv2-g+UniRA** | 8 | 8.6 | 5.22 | 119.7 | 0.69 | 0.66 |
| DINOv2-B+UniRA | 4 | 9.6 | 5.28 | 111.5 | 0.69 | 0.64 |
| DINOv2-B+UniRA | 6 | 8.7 | 5.33 | 116.4 | 0.69 | 0.65 |
| DINOv2-B+UniRA | 8 | 8.5 | 5.24 | 119.3 | 0.69 | 0.66 |
| DINOv2-B+UniRA | 10 | 9.1 | 5.34 | 112.3 | 0.69 | 0.65 |
| DINOv2-B+UniRA | 12 | 9.9 | 5.15 | 110.8 | 0.70 | 0.64 |
| DINOv2-B+UniRA | 14 | 10.4 | 5.14 | 107.6 | 0.69 | 0.64 |
| DINOv2-B+UniRA | 16 | 11.2 | 5.17 | 102.4 | 0.69 | 0.64 |

once. This strategy stabilizes adversarial training and prevents the generator from diverging in the early stages due to an overly strong discriminator.

## C EFFECT OF DIFFERENT PRETRAINED ENCODERS

As shown in Table A2, UniRA consistently improves generation quality across various pretrained encoders, outperforming both the original model (SiT) and REPA in terms of FID scores. This demonstrates that our approach effectively leverages pretrained representations for enhanced synthesis. Next, we evaluate the impact of encoder size by comparing different DINOv2 variants. Consistent with observations in REPA, we find that increasing the encoder size cannot leads to marginal performance improvements. This suggests that UniRA primarily benefits from the structural alignment of features rather than the absolute model capacity. Finally, we examine the importance of representation extraction depth and observe that aligning only the early-layer representations of the pretrained encoder is sufficient to achieve strong performance. This finding indicates that lower-level representations contain enough information for effective alignment, reducing the need for deeper feature supervision. These results highlight the robustness of UniRA across different pretrained encoder configurations and suggest that carefully selecting the alignment depth can improve efficiency without compromising quality.

## D DETAILS OF REPRESENTATION ANALYSES

**Structural organization of features.** For each image, we extract intermediate features at timestep $t = 0.5$ from the 8th transformer block of different denoising models (SiT, REPA, UniRA). For DINOv2, we instead use the output of the final layer. From each feature map, one spatial embedding is selected (marked with a red cross in the visualizations), and its cosine similarity with all other embeddings in the same map is computed. The resulting similarity map serves as the structural heatmap for that image. In the main text, we present six representative cases for visualization.

Table A3: Ablation study for alignment strategies with CFG($\omega = 2.0$)

| + Local | + Struc | + Global | FID↓ | IS↑ | Training Speed(step/s)↑ |
|---|---|---|---|---|---|
| ✓ | | | 1.96 | 264.0 | **2.41** |
| ✓ | ✓ | | 1.84 | 273.5 | 2.40 |
| ✓ | | ✓ | 1.90 | 270.4 | 1.78 |
| | ✓ | ✓ | 2.03 | 263.4 | 1.78 |
| ✓ | ✓ | ✓ | **1.82** | **279.8** | 1.76 |

**Semantic predictability and generation quality.** We follow the linear probing setup from REPA. A parameter-free batch normalization layer is applied before the linear classifier, which is trained for 90 epochs with a batch size of 16,384. We use the Adam optimizer with a cosine learning rate decay schedule, starting from an initial learning rate of 0.001. Features are extracted from the 8th transformer block at timestep $t = 0$ for both REPA and UniRA. Probe accuracy is then correlated with the FID scores obtained from the same checkpoints.

**Layer-wise analysis.** We extract intermediate features from different transformer blocks of the denoiser at timestep $t = 0.5$. Linear probes are trained using the same protocol as in the semantic predictability analysis. Accuracy is reported as a function of the layer index, illustrating how representational quality evolves across network depth.

**Timestep robustness.** To analyze robustness across diffusion timesteps, we extract features from the 8th transformer block at $t = 0$, $t = 0.25$, and $t = 0.5$. Probes are trained using the same configuration as in the semantic predictability analysis. This enables comparison of semantic predictability under varying levels of input corruption.

**Redundancy in representations.** We randomly sample 10,000 validation images and extract intermediate features from the 8th transformer block at timestep $t = 0.5$. For each model, the feature covariance matrix is computed, and PCA is applied to obtain the explained variance ratios. The final curve is obtained by averaging the cumulative explained variance ratios across all samples, where a lower curve indicates reduced redundancy and richer feature diversity.

# E  TRAINING OBJECTIVE OF DENOISEING MODEL

A transformer-based network $f_\theta$ refines noisy latent representations into high-quality image representations. Similar to REPA, our diffusion network is trained with a velocity prediction objective to enhance the generative process. Given a noisy latent representation $x_t$ at timestep $t$, the goal is to predict the velocity of the clean data $v$, defined as:

$$v_t = \frac{x_t - x_0}{\sigma_t} \tag{A1}$$

where $x_0$ is the clean latent representation, and $\sigma_t$ is the noise level at timestep $t$. The diffusion network is optimized to minimize the velocity prediction loss:

$$\mathcal{L}_{velocity}(\theta) = \mathbb{E}_{t,x_0,\epsilon}[||f_\theta(x_t, t) - v_t)||_2^2] \tag{A2}$$

where $\epsilon \sim \mathcal{N}(0, I)$ represents Gaussian noise. This objective ensures that the diffusion network learns an effective generative process for reconstructing image representations from noisy inputs.

# F  MORE ABLATIONS

**Effect of Alignment Strategies with CFG.** To complement Table 4, we report ablations under classifier-free guidance (CFG) in Table A3. Using the same sampling setup as the main experiments, we observe that CFG improves absolute FID/IS for all models, but the relative trend remains unchanged: UniRA with all three alignment components performs best, and removing any component leads to predictable degradation. This confirms that UniRA's gains stem from improved internal representations rather than sampling-specific benefits.

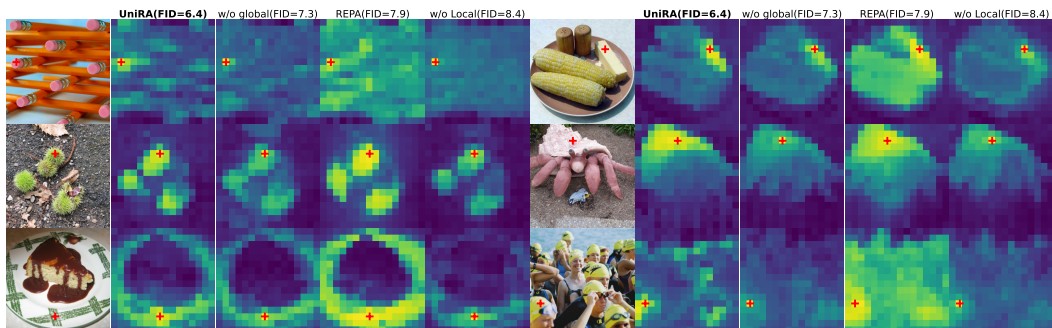

Figure A1: Effect of removing local or global alignment on structural correlation maps.

**Computing Resources.** All models are trained on 8×A100 (80GB) GPUs with fully sharded data parallelism. Table A3 includes the measured training throughput (steps/s) for different alignment combinations. Local alignment alone is fastest (2.41 steps/s), and adding structural alignment has negligible cost (2.40 steps/s). The global alignment term is the main source of overhead, reducing throughput to 1.76 steps/s due to discriminator updates. As discussed in Section 3.4, UniRA's primary gains come from the local and structural components; the global term is included as an optional refinement that provides small additional improvements at the cost of slower training.

**Structural Correlation Under Alignment Ablations.** We further visualize the structural similarity maps of UniRA and its ablations. Following Figure 5, features are extracted from the 8th transformer block at $t = 0.5$, and the cosine similarity between a reference patch token and all other tokens is computed. As shown in Figure A1, removing global alignment yields only mild changes, consistent with its role as a global variance refinement step. In contrast, removing local alignment significantly disrupts spatial coherence: heatmaps become noisier, semantic locality weakens, and relational structure degrades. These visual findings support the statement in the main paper that global coherence depends on stable local semantics, whereas removing the global term does not produce comparable degradation.

# G Limitations And Future Work

While UniRA demonstrates strong performance and consistently improves image generation quality, there remain natural directions for further exploration.

**Reliance on pretrained encoders.** Our framework leverages frozen vision encoders as semantic references, which may introduce mismatches when the target domain differs significantly. A promising direction is to explore adaptive or jointly optimized encoders that reduce dependence on external models.

**Balancing multiple objectives.** The method involves several alignment terms whose relative weighting influences training dynamics. Although our experiments show that UniRA is robust within reasonable ranges, automated or adaptive strategies for balancing objectives could further enhance stability and reduce manual tuning.

**Extension to broader domains.** Finally, while we focused on image generation, the unified representation alignment principle naturally extends to more challenging domains such as high-resolution synthesis, video, or 3D content, offering rich opportunities for future research.

# H QUALITATIVE RESULTS

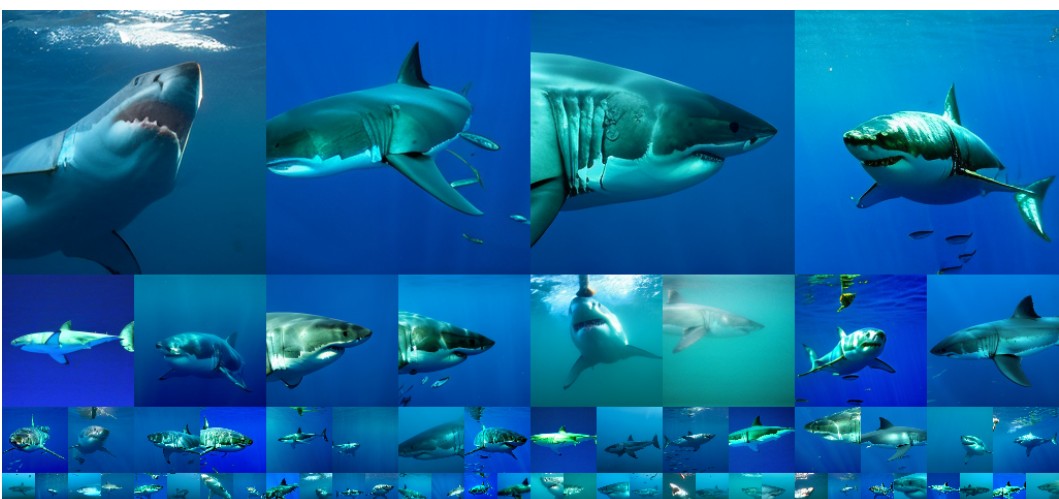

Figure A2: Uncurated generation results of SiT-XL/2+REPA. We use classifier-free guidance with $w = 4.0$. Class label="great white shark"(2).

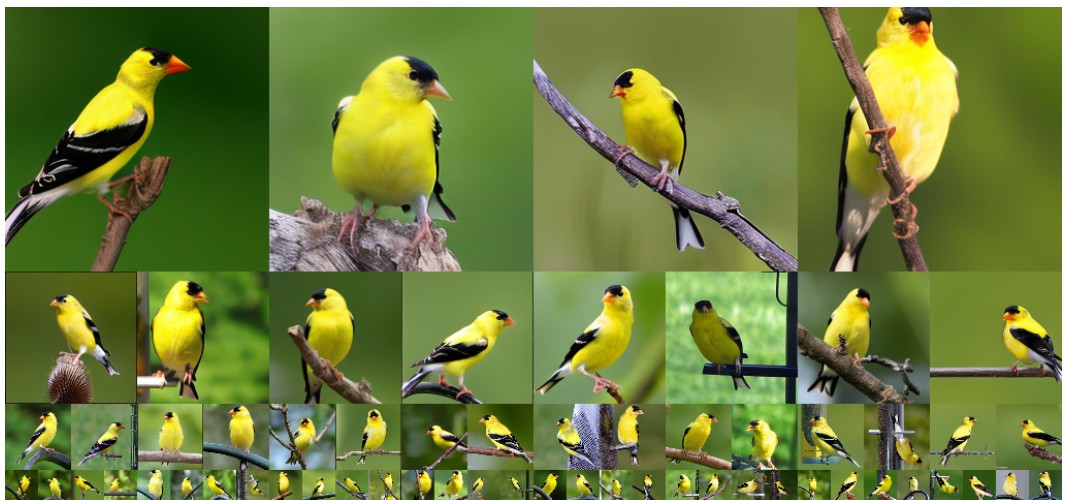

Figure A3: Uncurated generation results of SiT-XL/2+REPA. We use classifier-free guidance with $w = 4.0$. Class label="goldfinch"(11).

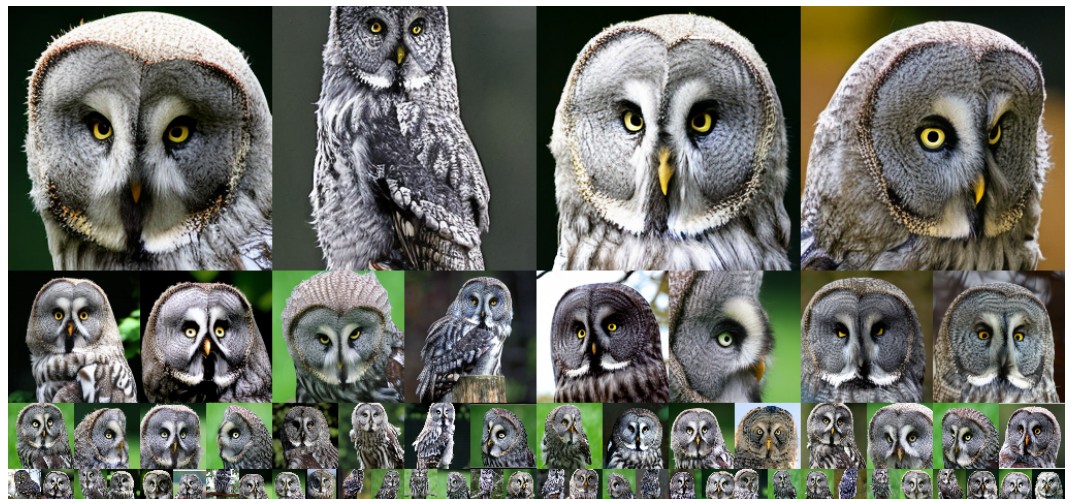

Figure A4: Uncurated generation results of SiT-XL/2+REPA. We use classifier-free guidance with $w = 4.0$. Class label="great grey owl"(24).

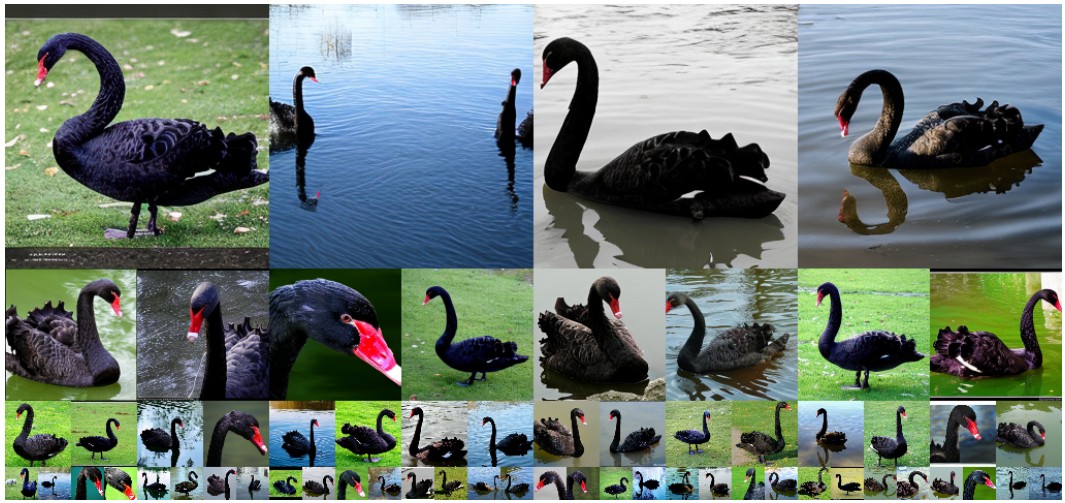

Figure A5: Uncurated generation results of SiT-XL/2+REPA. We use classifier-free guidance with $w = 4.0$. Class label="black swan"(100).

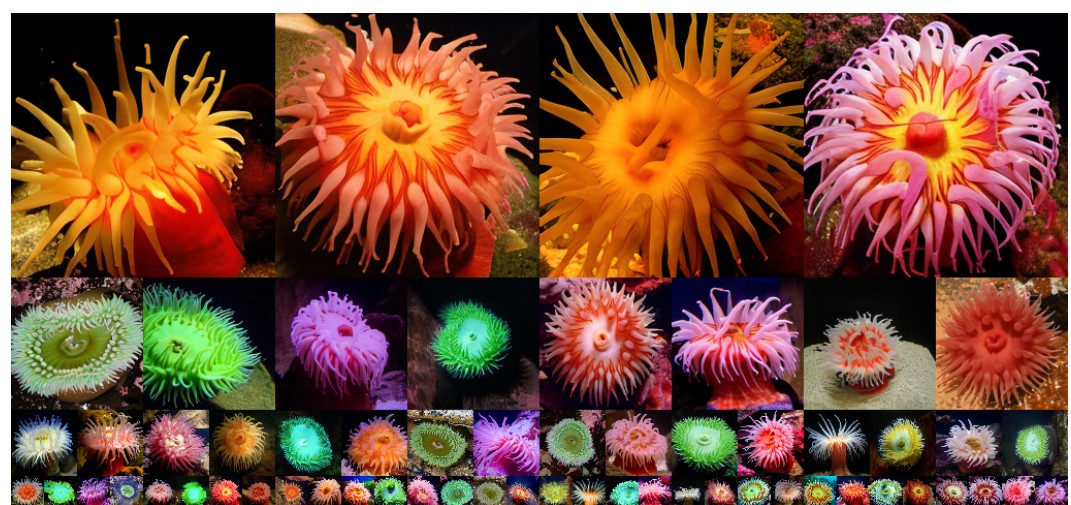

Figure A6: Uncurated generation results of SiT-XL/2+REPA. We use classifier-free guidance with $w = 4.0$. Class label="sea anemone"(108).

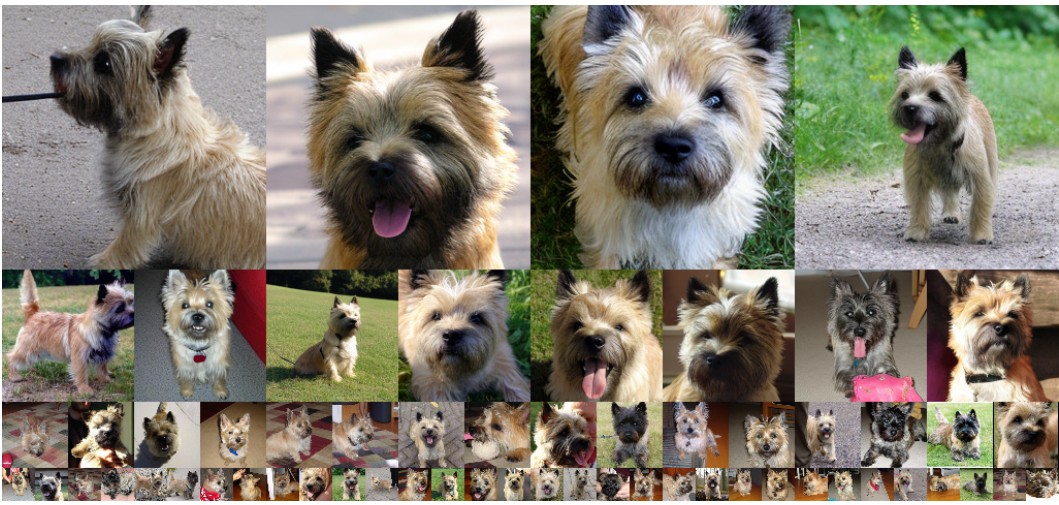

Figure A7: Uncurated generation results of SiT-XL/2+REPA. We use classifier-free guidance with $w = 4.0$. Class label="cairn"(192).

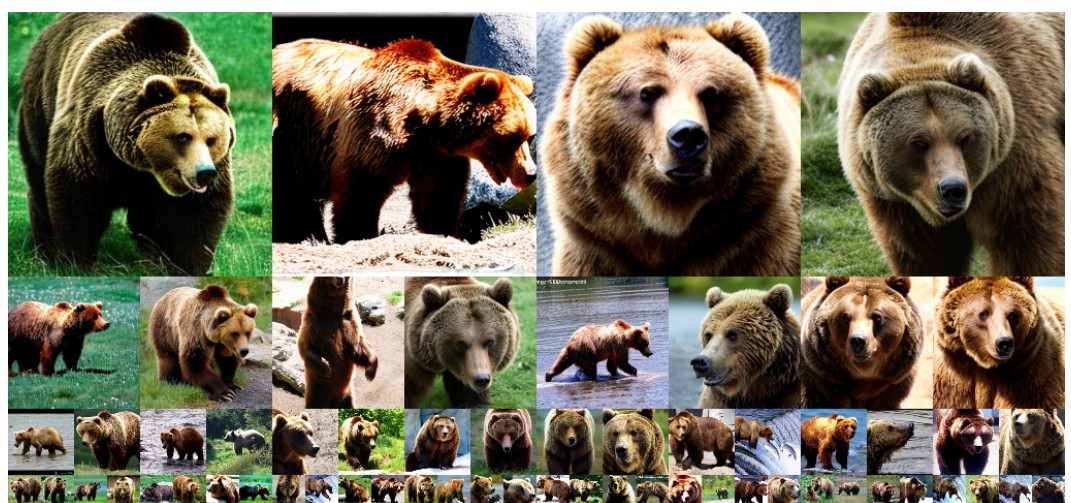

Figure A8: Uncurated generation results of SiT-XL/2+REPA. We use classifier-free guidance with $w = 4.0$. Class label="brown bear"(294).

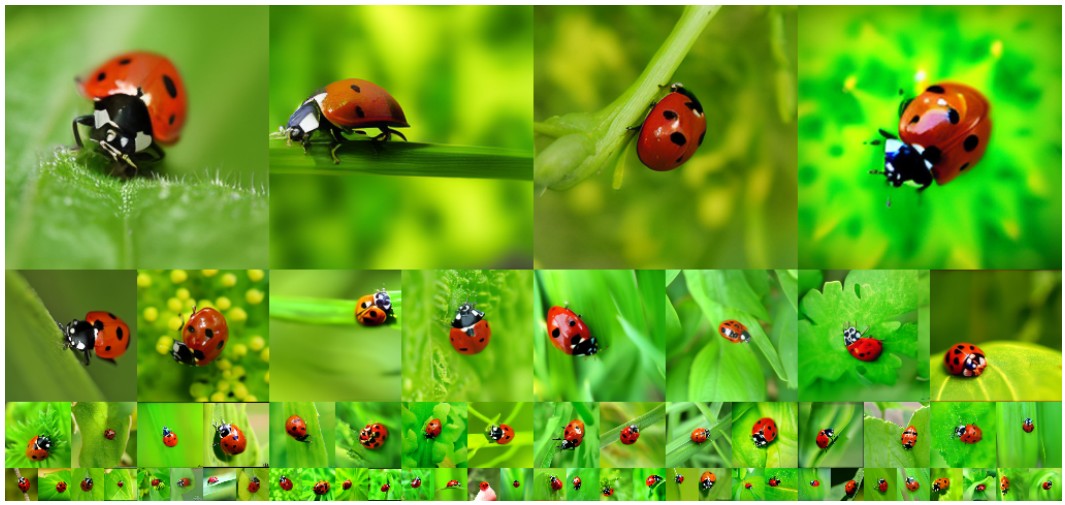

Figure A9: Uncurated generation results of SiT-XL/2+REPA. We use classifier-free guidance with $w = 4.0$. Class label="ladybug"(301).

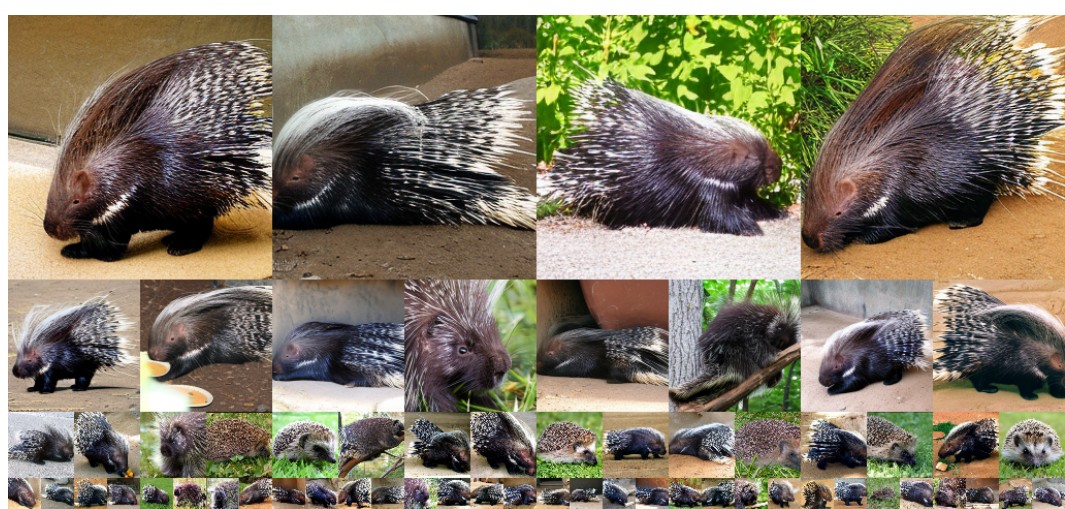

Figure A10: Uncurated generation results of SiT-XL/2+REPA. We use classifier-free guidance with $w = 4.0$. Class label="porcupine"(334).

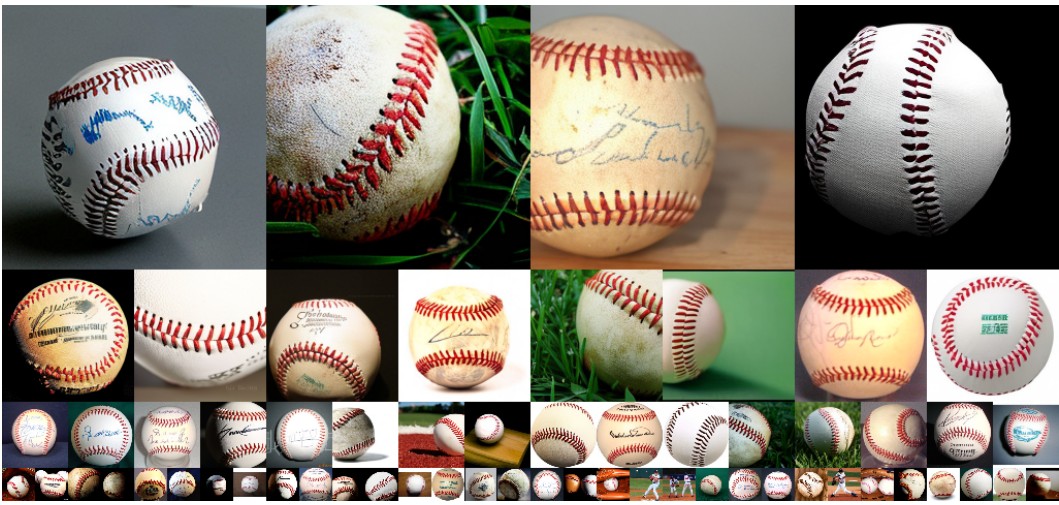

Figure A11: Uncurated generation results of SiT-XL/2+REPA. We use classifier-free guidance with $w = 4.0$. Class label="baseball"(429).

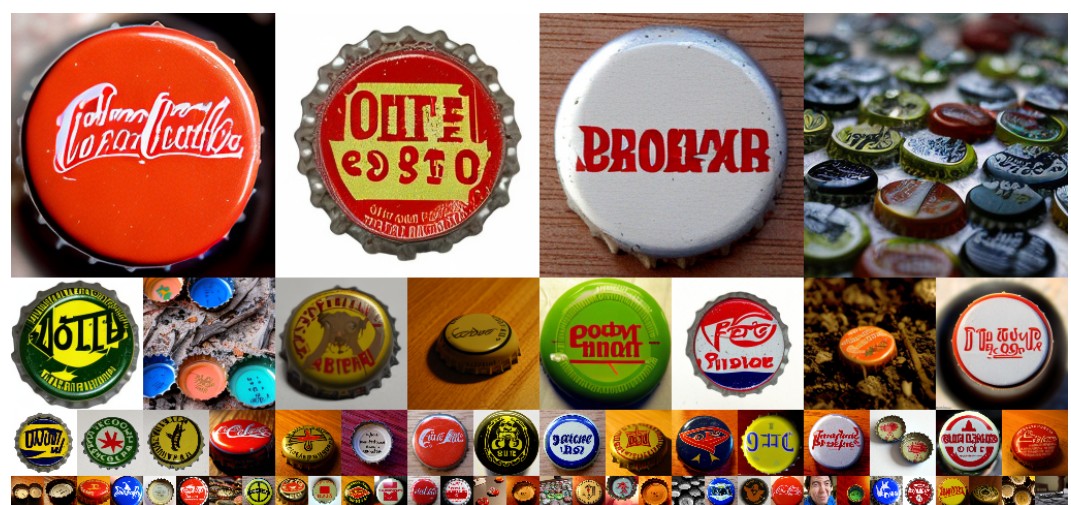

Figure A12: Uncurated generation results of SiT-XL/2+REPA. We use classifier-free guidance with $w = 4.0$. Class label="bottlecap"(455).

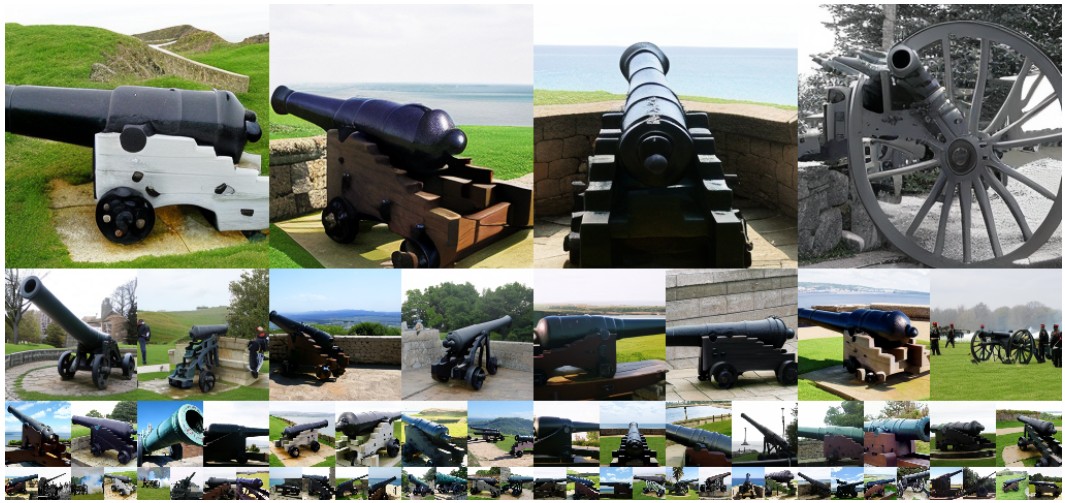

Figure A13: Uncurated generation results of SiT-XL/2+REPA. We use classifier-free guidance with $w = 4.0$. Class label="cannon"(471).

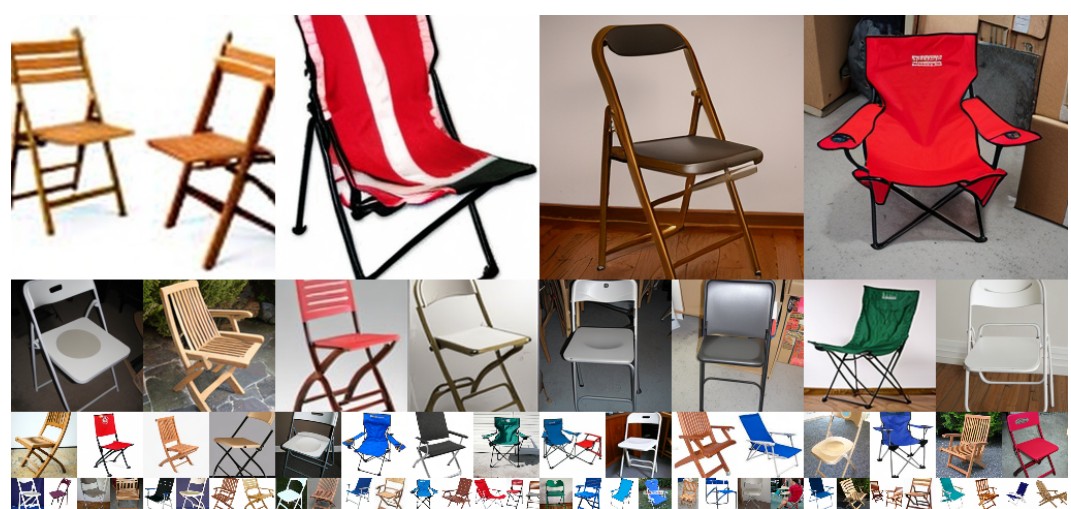

Figure A14: Uncurated generation results of SiT-XL/2+REPA. We use classifier-free guidance with $w = 4.0$. Class label="folding chair"(559).

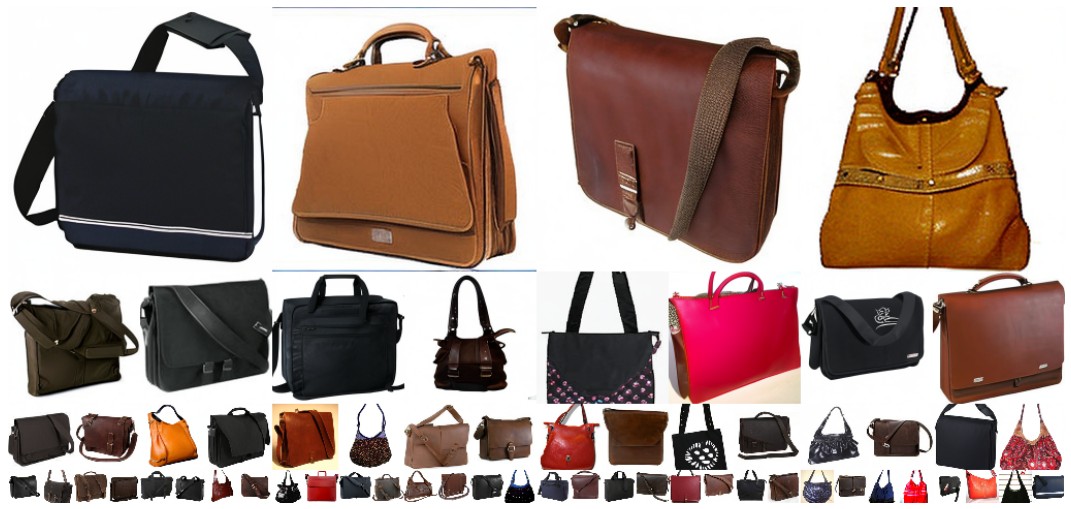

Figure A15: Uncurated generation results of SiT-XL/2+REPA. We use classifier-free guidance with $w = 4.0$. Class label="mailbag"(636).

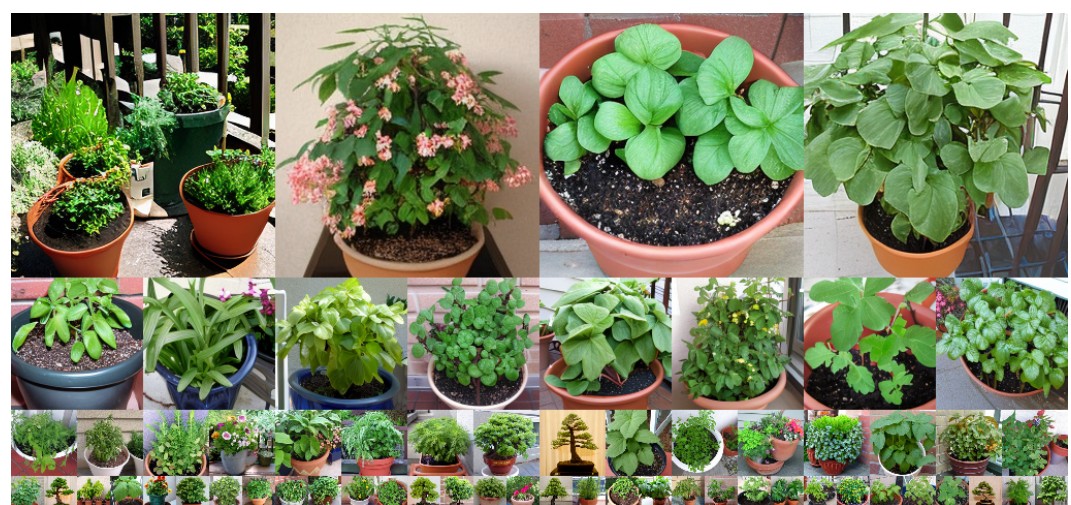

Figure A16: Uncurated generation results of SiT-XL/2+REPA. We use classifier-free guidance with $w = 4.0$. Class label="pot"(738).

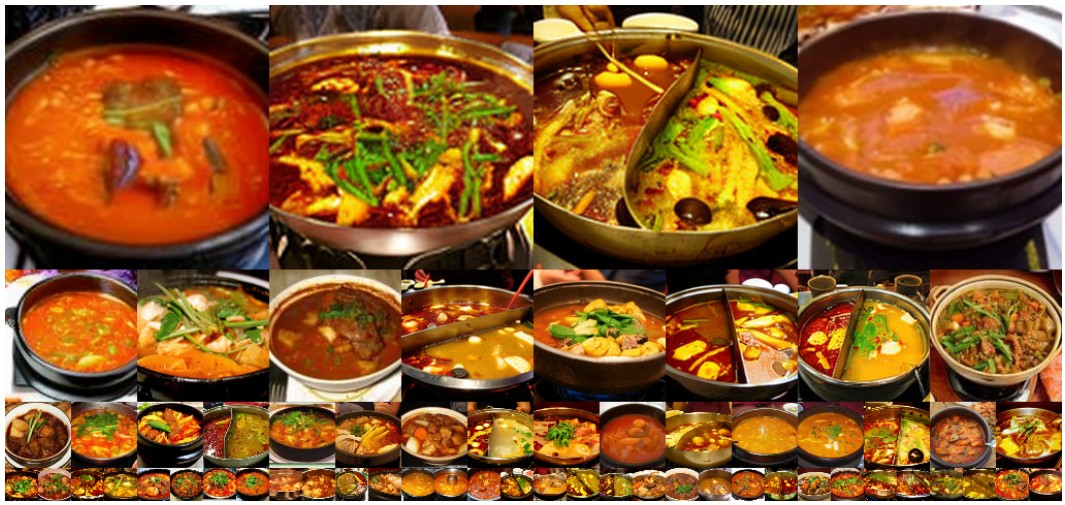

Figure A17: Uncurated generation results of SiT-XL/2+REPA. We use classifier-free guidance with $w = 4.0$. Class label="hot pot"(926).

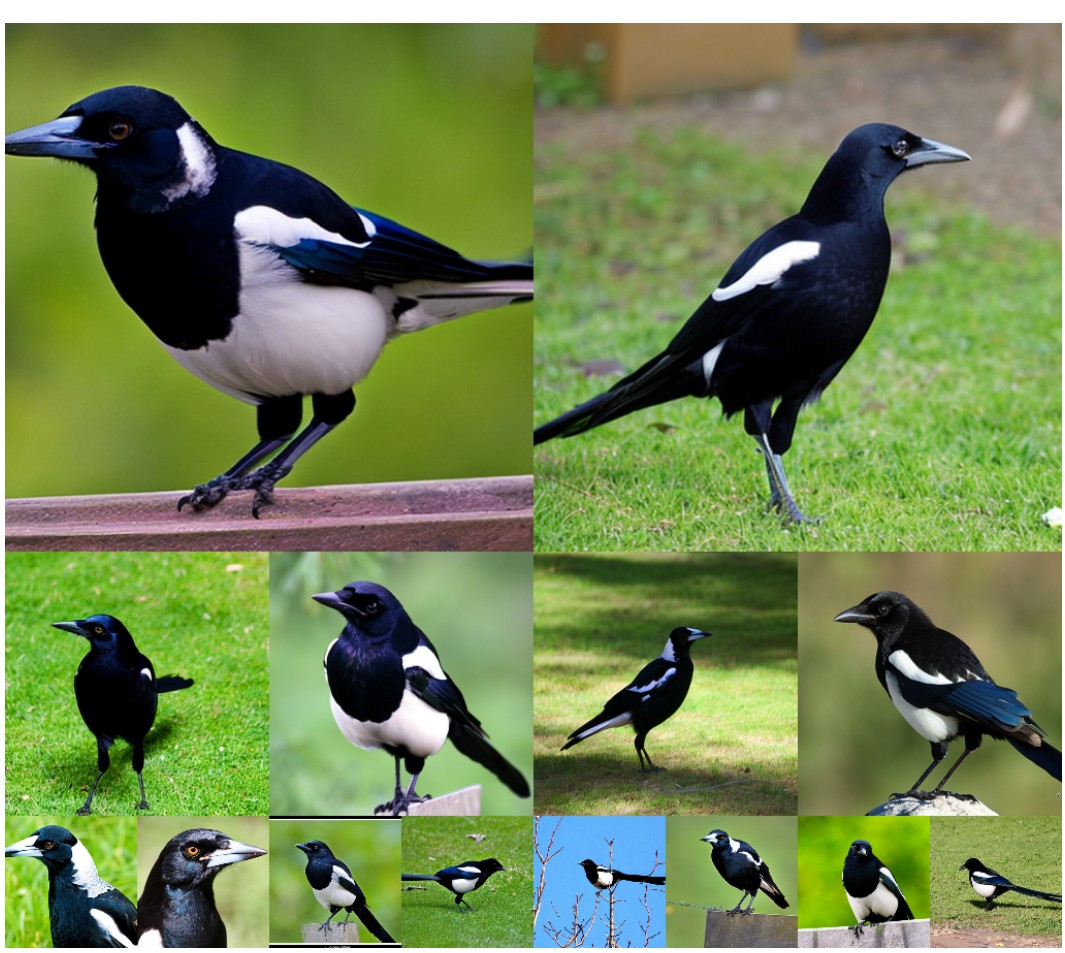

Figure A18: Uncurated $512 \times 512$ generation results of SiT-XL/2+REPA. We use classifier-free guidance with $w = 4.0$. Class label="magpie"(18).

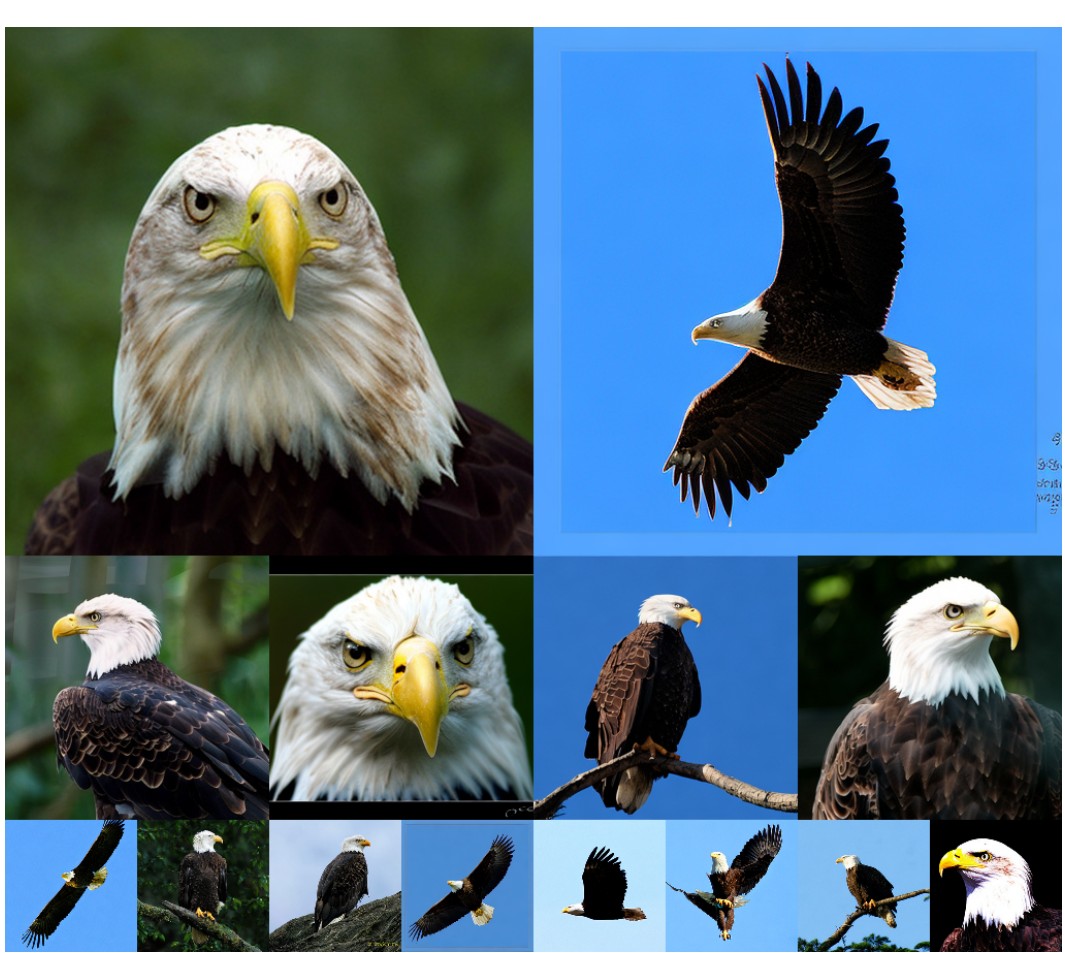

Figure A19: Uncurated $512\times512$ generation results of SiT-XL/2+REPA. We use classifier-free guidance with $w = 4.0$. Class label="bald eagle"(22).

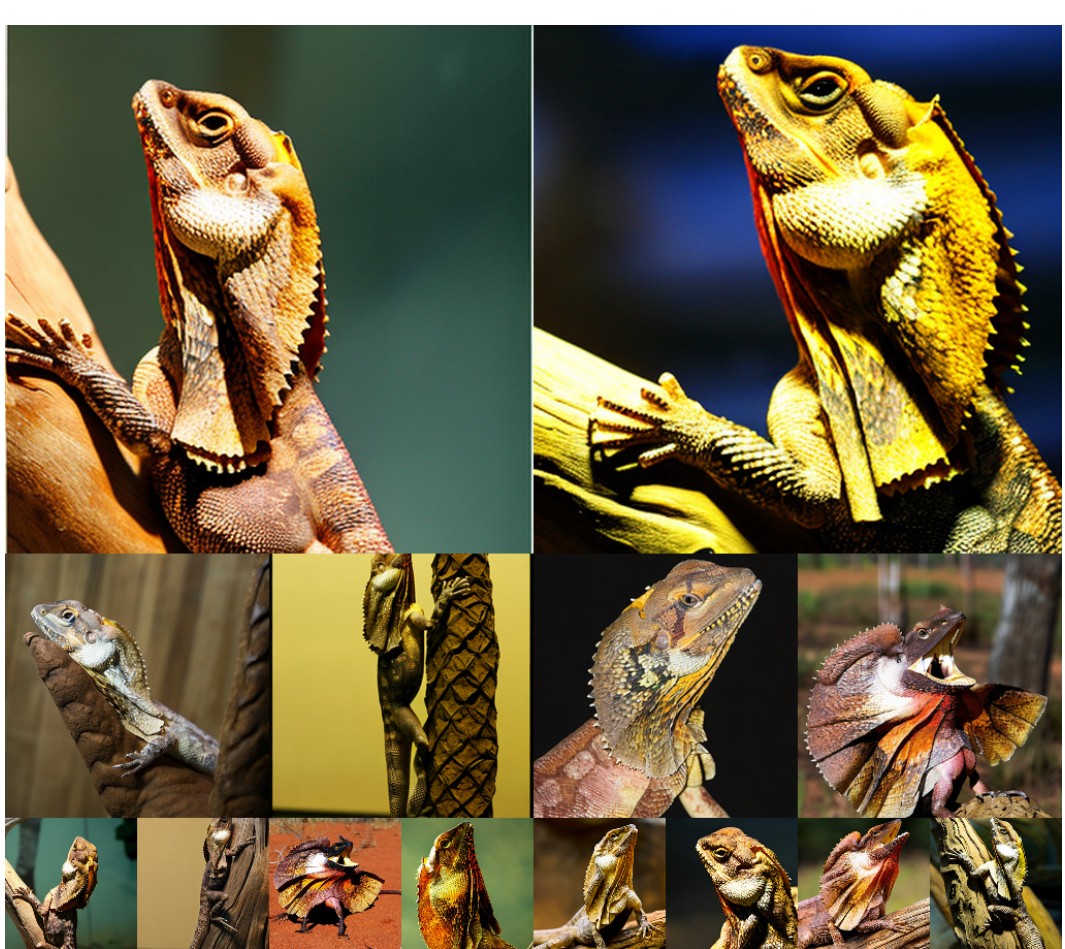

Figure A20: Uncurated $512 \times 512$ generation results of SiT-XL/2+REPA. We use classifier-free guidance with $w = 4.0$. Class label="frilled lizard"(43).

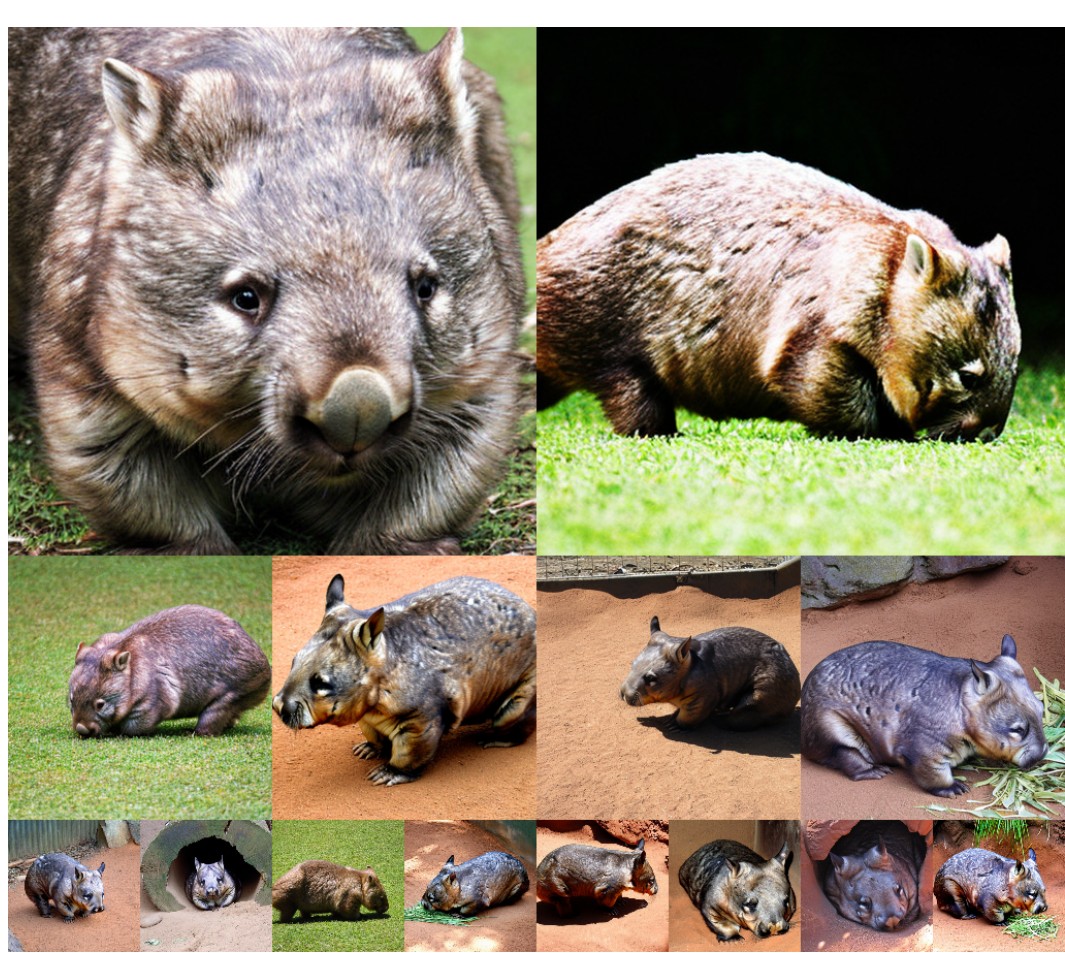

Figure A21: Uncurated $512{\times}512$ generation results of SiT-XL/2+REPA. We use classifier-free guidance with $w = 4.0$. Class label="wombat"(106).

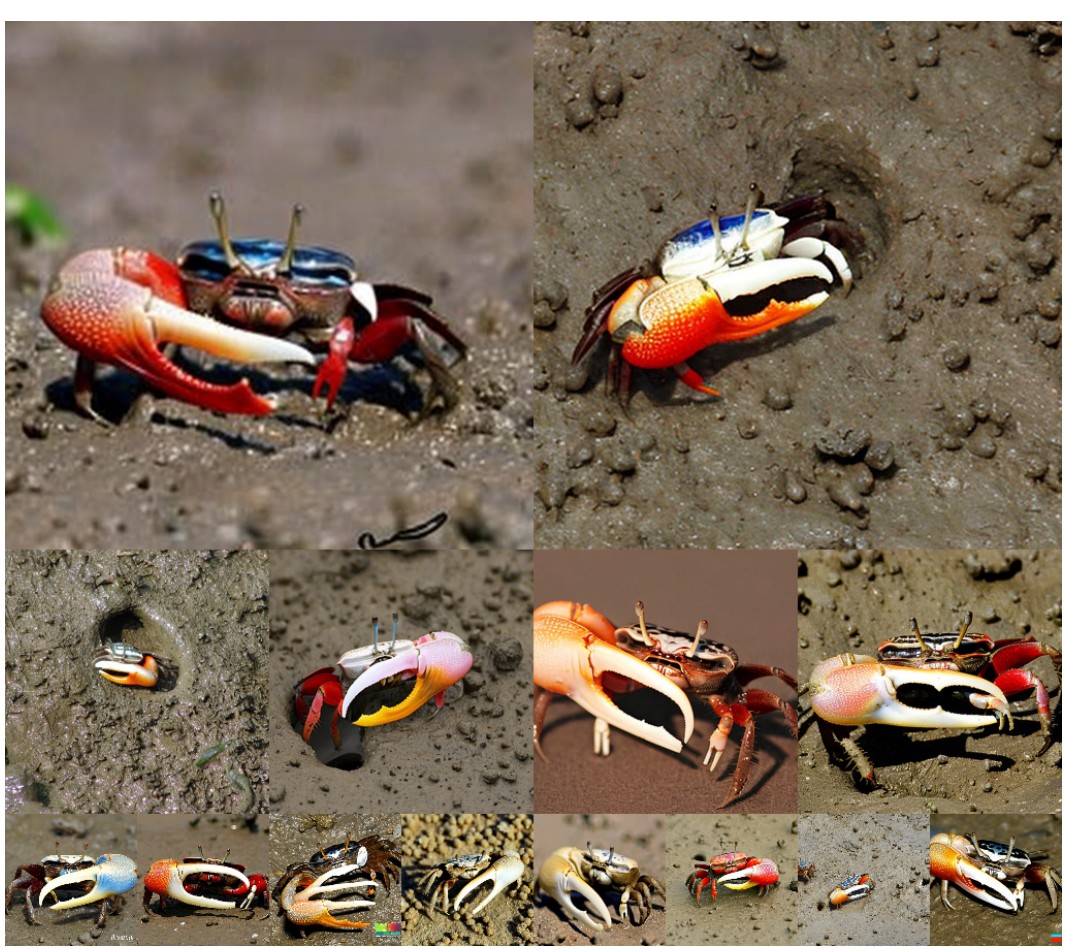

Figure A22: Uncurated $512 \times 512$ generation results of SiT-XL/2+REPA. We use classifier-free guidance with $w = 4.0$. Class label="fiddler crab"(120).

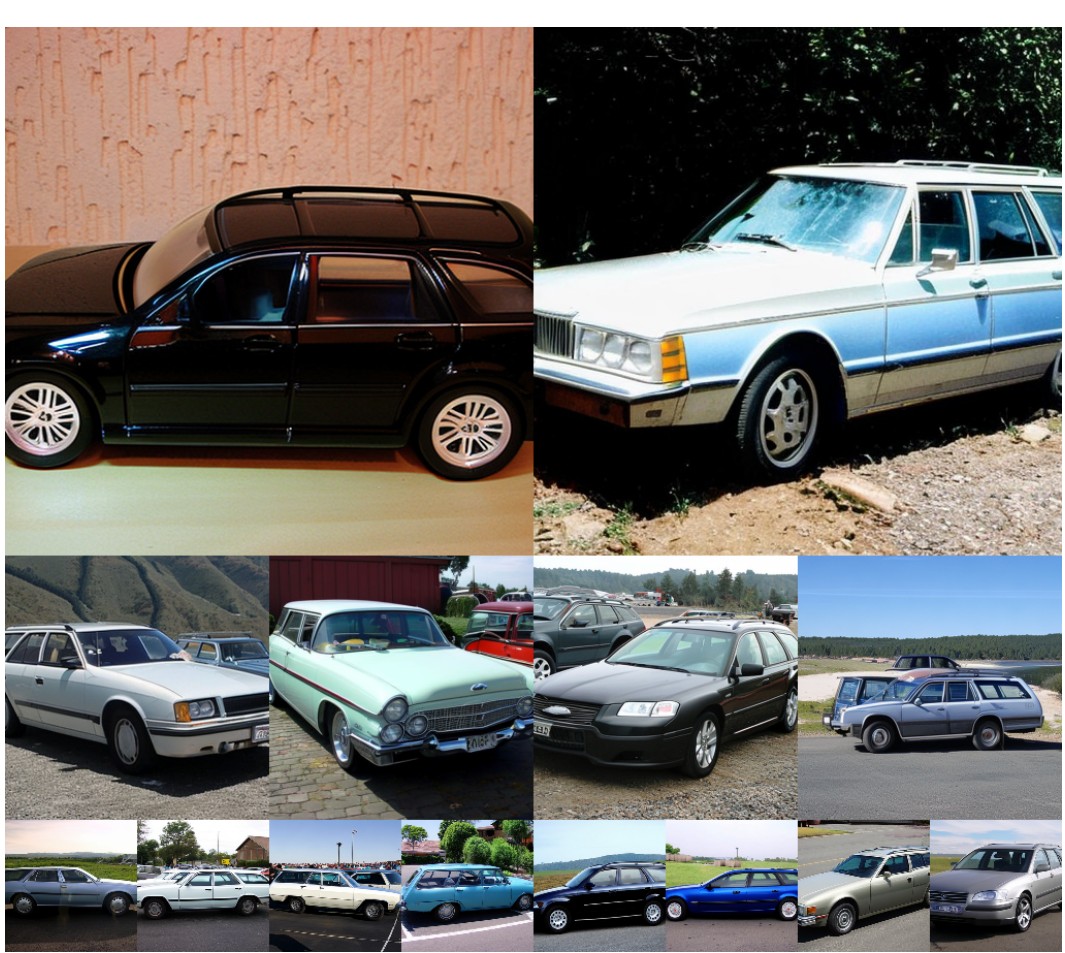

Figure A23: Uncurated $512 \times 512$ generation results of SiT-XL/2+REPA. We use classifier-free guidance with $w = 4.0$. Class label="beach wagon"(436).

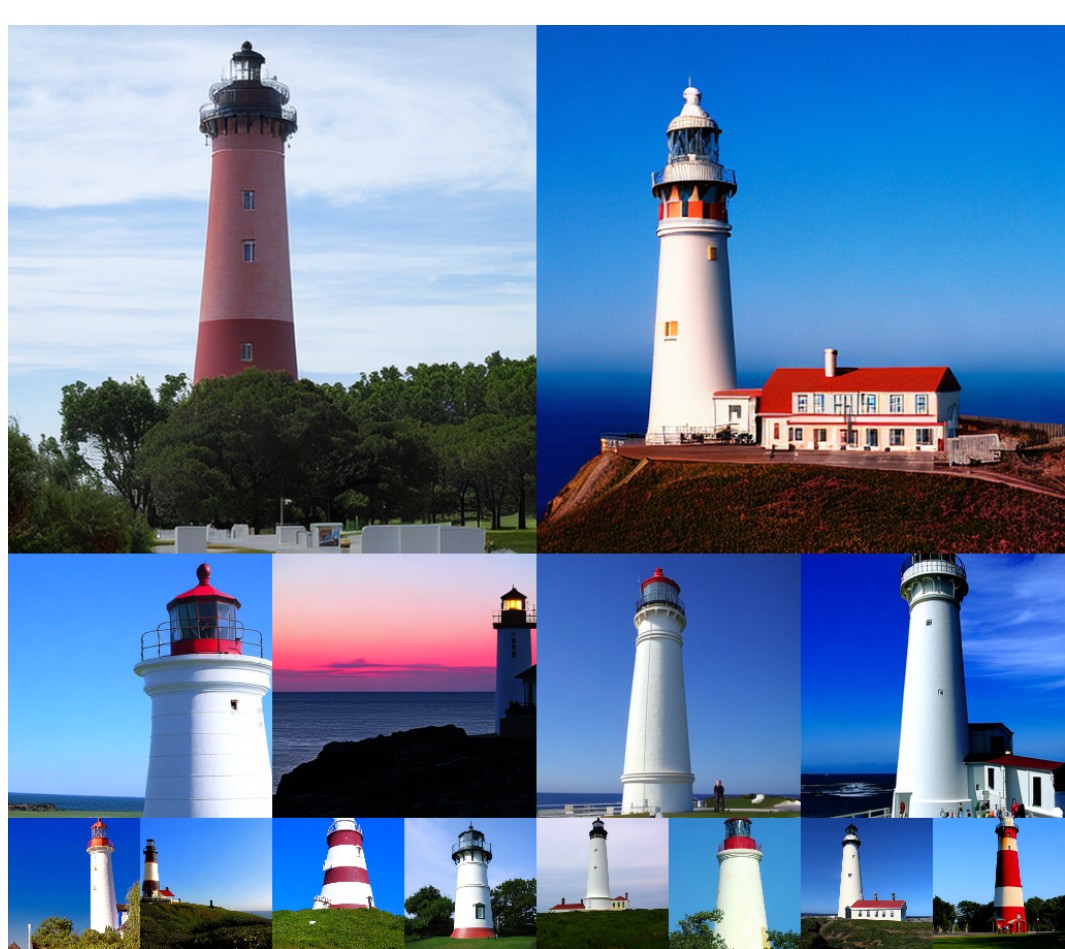

Figure A24: Uncurated $512 \times 512$ generation results of SiT-XL/2+REPA. We use classifier-free guidance with $w = 4.0$. Class label="beacon"(437).

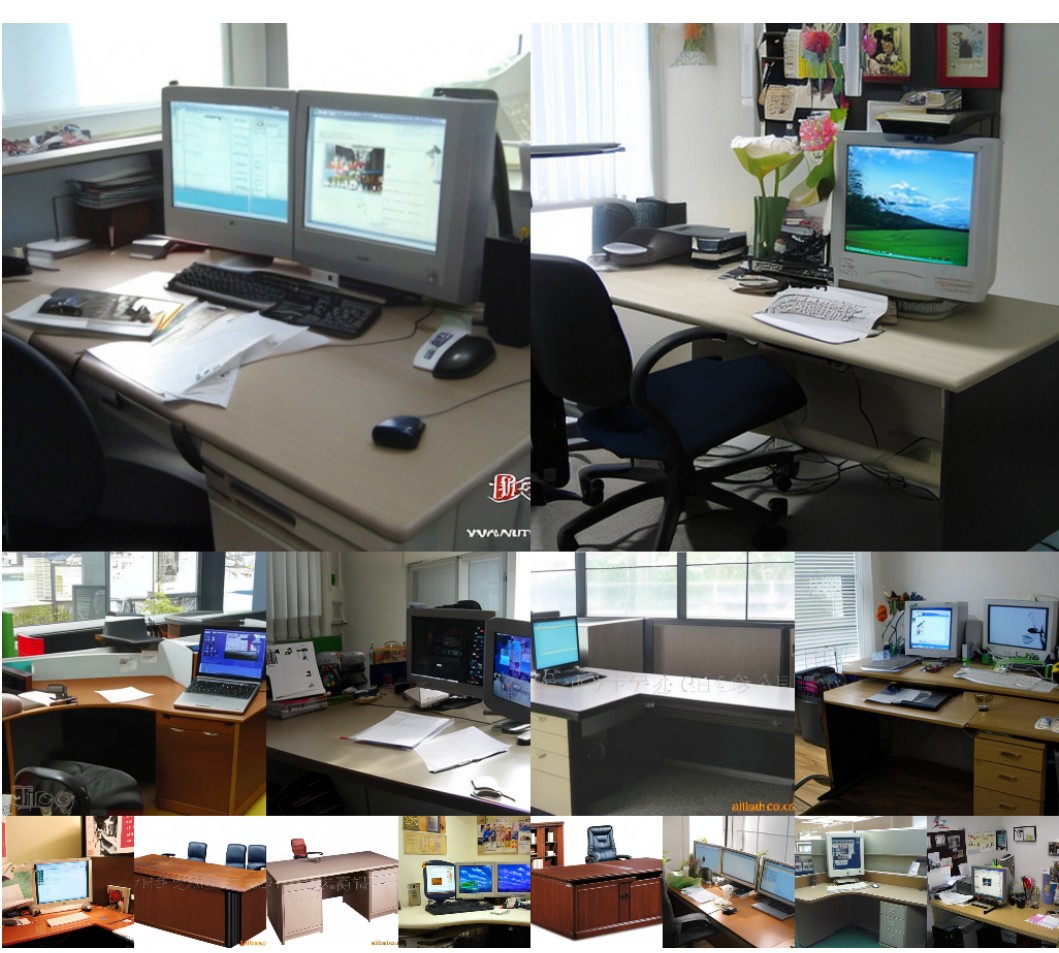

Figure A25: Uncurated $512 \times 512$ generation results of SiT-XL/2+REPA. We use classifier-free guidance with $w = 4.0$. Class label="desk"(526).

