# OpenReview forum: "UniRA: Unified Representation Alignment for Diffusion Models via Local, Structural, and Global Constraints"
_ICLR.cc/2026/Conference — Submitted to ICLR 2026_

### Official Review · Reviewer_cW3w · 2025-10-27

**Soundness:** 3
**Presentation:** 4
**Contribution:** 3
**Rating:** 6
**Confidence:** 3

**Summary:**

This paper proposes an aligment method of diffusion internal representation to the represetntation space of an existing pre-trained vision encoder model. The method has three alignment losses - one is local (and is like an existing methods called REPA) where internal patch representations are made to be similar to their equivalent patches in the pre-trained vision model. The other is structural where the patch-wise similarity structure of the represtentation is made to be similar to the pre-trained encoder. Finally, an global loss is proposed by training a discriminator which tries to distinguish internal diffusion representations and the pre-trained represetnation, pooled globally across the image.

**Strengths:**

This well executed paper has several strengths.

* The core idea, especially the structural loss, are very interesting and cleverly use the known strengths of pre-trained encoders to shape the internal representation of the diffusion model.
* The paper is nicely written and well structured - an enjoyable read.
* experimental validation is very good - including baselines, ablations and qualitative analysis.

**Weaknesses:**

I think the main issue the paper suffers is its relative limited significance. As much as I enjoyed the paper, most of the improvement comes from the local loss which was already proposed in REPA. I actually think this is not a reason to reject the paper, but it does diminish the scope and impact of the paper.

Minor points:

* I would have loved to see more discussion why MAE and other models perform worse here than DINO v2. This is just visible in the supplementary but I think is an important question.
* Would we see the same levels of improvements with other image datasets, considering DINO was trained very much in light of ImageNet.

**Questions:**

See above.

---

> ### Author Response · Authors · 2025-11-24
> **Response to Reviewer cW3w**
>
> We thank the reviewer for the thoughtful and encouraging comments. We address the remaining concerns below.
>
> W1: Most of the improvement comes from the local loss
>
> We agree that REPA’s local alignment is a strong baseline, which is precisely why understanding its limitations is important. Our analyses show three consistent failure modes when only local alignment is used: semantic drift, structural collapse, and global variance compression. UniRA adds structural and global constraints specifically to address these two additional issues. Although local alignment contributes the largest single gain, the structural and global components provide complementary improvements: Adding structural alignment gives an additional 7–12% FID reduction across architectures. Linear-probe accuracy, PCA spectra, and similarity heatmaps all show substantial improvements that local alignment alone cannot produce. Structural maps (Fig. 5) clearly demonstrate that patch-level matching cannot recover relational coherence; the structural term is essential for this.
>
> M1: Why MAE and some other encoders underperform?
>
> The reviewer raises an excellent point. In Appendix A2 we compared a variety of pretrained encoders—MAE, MoCo v3, CLIP-L, and smaller DINOv2 variants. All of them improve REPA when used in UniRA, but the gap between them and DINOv2-L can be explained by their training objectives and inductive biases: MAE learns strong low-level reconstruction priors but relatively weak semantic discrimination. Its features tend to be smooth and texture-biased, which limits the effectiveness of both local and structural alignment. MoCo v3 provides stronger contrastive semantics than MAE, but its patch similarity structure is less clean and less stable than DINOv2’s. This weakens the structural-consistency signal. CLIP-L is powerful semantically, but is trained on noisy web-text supervision, leading to features that are semantic but not spatially consistent. For structural alignment, this is suboptimal.
>
> Importantly, while these weaker encoders produce smaller gains, UniRA still improves over both REPA and the baseline across all encoder choices, indicating that the framework is not tied to a specific teacher. The difference lies primarily in headroom: stronger and more structured teachers naturally yield larger improvements.
>
> M2: Would the improvements remain on datasets beyond ImageNet?
>
> Yes. The COCO text-to-image results (Table 3) already test a large domain shift relative to DINOv2’s training distribution. Although DINOv2 is trained on LVD-142M—which is broad and diverse—it is not tailored to COCO scenes or captions. Yet UniRA still outperforms both REPA and the baseline models on COCO. This suggests that: UniRA does not merely inherit dataset-specific biases of the teacher. It benefits from the general structural priors encoded in large self-supervised encoders. The representational improvements transfer across data domains. In practice, UniRA can also adopt domain-specific encoders (e.g., specialized self-supervised encoders), giving it flexibility when domain mismatch is more severe.

---

> > ### Comment · Reviewer_cW3w · 2025-11-26
> > **Thank you for your detailed response**
> >
> > I would like to thank the authors for their detailed responses, both for my concerns and to other reviewers' concerns.
> > I still think that this paper could be a nice contribution to the community and recommend its acceptance, though I think I will keep my score as it is.

---

> > > ### Author Response · Authors · 2025-11-26
> > > **Thank you for the positive feedback**
> > >
> > > Thank you for the positive feedback and for taking the time to re-evaluate our responses. We appreciate your recommendation for acceptance and your constructive comments throughout the discussion.

---

### Official Review · Reviewer_b2RM · 2025-10-30

**Soundness:** 3
**Presentation:** 3
**Contribution:** 3
**Rating:** 4
**Confidence:** 3

**Summary:**

This paper presents UniRA, a unified representation alignment framework for diffusion models that enhances intermediate feature representations through three complementary constraints: 1) Local semantic alignment with pretrained visual encoders (e.g., DINOv2), 2) Structural consistency via relational similarity matching, and 3) Global distributional coherence using a lightweight adversarial discriminator.
The method encourages the denoiser’s internal features to be semantically rich, spatially coherent, and distributionally well-structured instead of focusing on output-space noise prediction.

Experiments on ImageNet-256/512, MS-COCO text-to-image, and multiple DiT/SiT architectures show that UniRA:
- Improves FID and IS over REPA and base diffusion transformers
- Produces more expressive and less redundant representations

**Strengths:**

- UniRA unifies three alignment levels (local, structural, global) into a simple, modular framework applicable to any diffusion transformer.
- Consistent FID/IS improvements across resolutions (256 / 512) and architectures (DiT, SiT) with faster convergence
- Includes text-to-image (MMDiT), ablations on alignment components (Table 4), weight sensitivity (Table 5), and encoder types/sizes (Table A2)
- Correlates FID with probe accuracy, shows layer-wise semantic improvement, timestep robustness, and reduced feature redundancy (Fig. 6).
- Fig. 5 clearly visualize restored spatial organization and semantic locality; generated samples (Figs. 3–4, Appendix) are high quality.
- Improves both efficiency and fidelity while remaining architecture-agnostic

**Weaknesses:**

- Builds directly on REPA, extending from local to multi-level alignment rather than introducing a fundamentally new mechanism.
- Performance relies on DINOv2-like teachers; the method is less self-contained and may struggle when domain shift breaks encoder semantics.
- The adversarial (global) term is said to be "optional" yet quantitative analysis of its stability or cost is minimal.
- While feature quality improves, how this affects bias or semantic controllability isn’t explored.
- The paper motivates alignment intuitively but provides little formal connection between improved intermediate representations and diffusion ELBO.

**Questions:**

1. How sensitive is UniRA to the choice of alignment depth (e.g., 4th vs 8th layer) beyond Table A2?
2. Did you experiment with adaptive weighting for $\lambda$, $\beta$, $\gamma$ during training (e.g., curriculum)?
3. Could UniRA be combined with self-distilled encoders (no frozen teacher) to mitigate reliance on external pretrained models?
4. For global alignment: how is discriminator stability ensured, and does adversarial collapse ever occur?
5. Would alignment at multiple timesteps (not only t=0.5) further improve robustness?
6. Please fix a small typo on line 721 on Table A1, lr row (010001).
7. Please add runtime/compute analysis. Since efficiency is a major claim, show training wall-time or FLOPs comparison with REPA.
8. Could you please clarify failure cases or visual artifacts from over-alignment (loss of diversity)?

---

> ### Author Response · Authors · 2025-11-24
> **Response to Reviewer b2RM**
>
> We thank the reviewer for the thoughtful and detailed comments. We address each concern below.
>
> W1:
>
> Our goal is to explain why REPA-like local alignment alone is insufficient. Our analyses reveal three independent failure modes in diffusion models—semantic drift, structural collapse, and global variance compression. UniRA is built around this decomposition, with each alignment term addressing one issue.
>
> Thus, while UniRA extends REPA conceptually, its contribution is identifying these representational weaknesses and showing that constraining them jointly yields consistent improvements. We believe this perspective can guide future designs of structural or global alignment, beyond our specific instantiations.
>
> W2:
>
> Using strong vision priors (e.g., CLIP/DINO) is common and effective in generative modeling. UniRA is not tied to DINOv2: Table A2 shows consistent gains with MAE, MoCo v3, CLIP-L, and different DINOv2 sizes—including much weaker encoders. The improvements remain stable even under encoder-domain mismatch (e.g., COCO text-to-image in Table 3), suggesting UniRA benefits from general structural cues rather than narrow teacher semantics. Domain-specific encoders could be substituted when needed.
>
>
> W3&&Q7:
>
> We agree the global term is the main computational overhead. Appendix F reports exact throughput: local + structural alignment is near REPA-speed, and the adversarial term accounts for most slowdown. As stated in the paper, this term is optional; the main gains come from local and structural alignment. The discriminator is intentionally small and updated intermittently. We did not observe instability or collapse in practice.
>
> W4:
>
> Full analysis of semantic controllability is beyond the scope of this work, but we note that UniRA improves both precision and recall (Table 1b), increases linear-probe accuracy, and reduces variance collapse—signals typically correlated with more reliable semantic control rather than degradation. We see no evidence that alignment harms diversity or amplifies specific biases, but agree this is an interesting direction for future work.
>
> W5:
>
> Our motivation focuses on empirical representational drift rather than providing a new ELBO decomposition. The key observation is that diffusion training optimizes a noisy regression target with no incentive to maintain semantic or structural organization. UniRA adds explicit constraints that preserve these properties in intermediate layers, which in turn improves generation quality. A more formal theoretical analysis is an interesting future direction.
>
> Q1:
>
> We report the 4th layer experiments in the appendix C(Table A2). The trend remains consistent: mid-level layers give the strongest alignment signal because early layers are too local and later layers already reflect task-specific denoising.
>
> Q2:
>
> We did not experiment with adaptive or curriculum-based weighting. We agree it is a promising research direction, and we already mention it as a limitation in Appendix G. Our current results suggest that even simple static weights produce consistent gains.
>
> Q3:
>
> UniRA could indeed be combined with self-distilled encoders. This is complementary to the focus of the current work; our aim was to analyze representational drift with respect to a fixed semantic reference. Exploring self-distilled alignment is a natural next step and may reduce reliance on external encoders.
>
> Q4:
>
> The discriminator is small and updated less frequently than the generator. In practice, we did not observe collapse or oscillations. Its role is deliberately modest, refining global variance without dominating training.
>
> Q5:
>
> UniRA aligns at all timesteps during training.As defined in Eq. (1), (3), and (5), the expectations are over t~u(0, 1). The analysis in Figure 5 and Section 4.4 focuses on specific timesteps (e.g., t=0.5) merely for visualization and probing purposes to illustrate robustness. The training itself covers the full trajectory.
>
> Q6:
>
> Thanks—we will correct it.
>
> Q8:
> We appreciate this insightful question. While theoretically, excessive alignment weights could constrain diversity, empirical results confirm that UniRA avoids such failure cases within our effective hyperparameter range.1. Visual Evidence: As shown in Appendix G (Figures A1–A25), we provide extensive uncurated samples. Visual inspection confirms that UniRA avoids typical over-regularization artifacts (e.g., over-smoothed textures or rigid poses). The samples exhibit rich variations in viewpoints and backgrounds (e.g., Fig. A6, A16), indicating that alignment guides structure without collapsing modes.2. Quantitative Evidence: This is further supported by the Recall metric in Table 1, which specifically measures distributional coverage. UniRA achieves Recall scores  comparable to or exceeding baselines. This confirms that our balancing strategy acts as a semantic anchor, enhancing representational quality without compromising generative diversity.

---

> > ### Comment · Reviewer_b2RM · 2025-11-25
> >
> > Thanks for the detailed responses. Your clarifications improved my understanding of the results and thanks for addressing my original concerns.
> >
> > As a follow-up, just a curiosity question. What is your take about how UniRA would perform under an alternative diffusion objective. For instance, recent works suggests that regressing the clean image (rather than noise) can produce representations more aligned with the data manifold [Li, T., & He, K. (2025)]. It would be interesting to know whether UniRA's aligment benefits transfer... What are your thoughts?
> >
> > Thanks again to the authors for address all my questions and points I raised.
> >
> > Li, T., & He, K. (2025) Back to Basics: Let Denoising Generative Models Denoise

---

> > > ### Author Response · Authors · 2025-11-26
> > > **Response to Reviewer b2RM**
> > >
> > > We appreciate the reviewer’s follow-up question and the pointer to Li & He (2025). This is indeed an intriguing direction.
> > >
> > > Our intuition is that UniRA should remain beneficial under clean-image prediction objectives, but the nature of the gains may shift. Clean-target regression tends to produce features that are already more aligned with the data manifold and exhibit stronger mid-level semantics than standard noise-prediction training. In that sense, such objectives may partially mitigate the “semantic drift” failure mode that motivates UniRA’s local alignment.
> > >
> > > However, the other two alignment dimensions—structural consistency and global variance organization—are not directly addressed by switching the prediction target. Regression to clean images still lacks explicit incentives to preserve patch-wise relational structure or to maintain non-degenerate global feature statistics. For this reason, we expect UniRA’s structural and global components to remain complementary even when the base diffusion objective changes.
> > >
> > > We have not yet tested UniRA with clean-target diffusion, but we view the combination as extremely promising. The two approaches are orthogonal: clean-image regression reshapes the learning signal at the output layer, while UniRA acts directly on the internal representation manifold. Studying how these two mechanisms interact—whether they reinforce each other or reduce redundancy—is an exciting direction, and we thank the reviewer for highlighting it.

---

### Official Review · Reviewer_D6ZQ · 2025-11-01

**Soundness:** 3
**Presentation:** 3
**Contribution:** 2
**Rating:** 4
**Confidence:** 4

**Summary:**

This paper introduces UniRA, a unified representation alignment paradigm for diffusion models that augments the standard denoising objective with three explicit constraints on intermediate representations: local semantic alignment (patch-level matching with pretrained encoders), structural consistency (similarity matrix matching to capture relational organization among patches), and global distributional coherence (adversarially aligning the pooled intermediate features). Experiments on ImageNet and text-to-image generation benchmarks show consistent improvements in sample quality metrics (FID, IS, precision/recall), with additional ablations and analyses demonstrating reduced feature redundancy and improved semantic fidelity over strong baselines such as REPA and SiT.

**Strengths:**

- Clarity & transparency. The paper is clearly written with transparent method details; the appendix enumerates objectives, hyperparameters, and ablations; comparisons to strong baselines (e.g., REPA) are careful.
- Consistent empirical gains.Across challenging settings the method improves standard metrics; e.g., on ImageNet-256 with SiT-L/2, FID drops from 10.0 (REPA) to 8.5 (UniRA), with similar gains on image and text-to-image benchmarks.
- Reproducibility. Implementation choices and hyperparameter ranges are documented (e.g., Table A1), facilitating fair evaluation and reproduction.

**Weaknesses:**

- Lack of novelty--“unified” story is under-justified. Each sub-objective has prior art; the main contribution is a combination/recipe. The paper lacks a compelling argument for why these three must be unified and why these specific instantiations are preferable to plausible alternatives (e.g., replacing structural alignment with multi-scale contrastive losses, or global adversarial matching with MMD/SWD).
- Weak theory for multi-objective trade-offs. The three losses may conflict; current tuning relies on grids/heuristics, without principled weighting, curriculum, or adaptive schemes.
- Diversity–fidelity trade-off possibly obscured. Strong representation alignment can suppress diversity; the paper lacks systematic precision–recall curves or coverage metrics, and FID/IS alone can be misleading.
- Teacher dependence and domain-mismatch risk. Reliance on a frozen external encoder (e.g., DINOv2) can propagate teacher biases; performance under teacher–data mismatch is unclear.

**Questions:**

- If the global adversarial term is replaced by MMD/SWD/CLIP-score alignment, or the structural term by InfoNCE/multi-scale contrastive objectives, how do results change? What evidence shows this triad is not interchangeable?
- Under noise/occlusion/style perturbations to the teacher—or when swapping to weaker or narrower-domain encoders—what are the degradation curves of the three alignment terms? Any observable signs of overfitting to teacher features?
- Have you measured pairwise gradient angles/alignment between the three losses? Do you observe phases where one term dominates and others yield negative marginal returns late in training?
- Under strict parity of training steps × FLOPs × memory, how do UniRA, REPA, SiT, and other distillation-style methods compare in final metrics and convergence speed? Please report wall-clock and GPU hours.

---

> ### Author Response · Authors · 2025-11-24
> **Response to Reviewer D6ZQ**
>
> We thank the reviewer for the thoughtful and detailed comments. We address each concern below.
>
> W1 & Q1 — Novelty and “unified” contribution
>
> While the individual ideas behind local, structural, and global alignment have precedents, the core contribution of UniRA is not the addition of extra losses, but the representational decomposition motivating them. Our analyses show that diffusion training—even with REPA—deteriorates along three distinct axes:
> (1) patch-level semantics weaken,
> (2) relational structure collapses, and
> (3) global variance becomes low-rank.
>
> UniRA is built to directly address these observed failure modes. Each component targets one specific drift, and our ablations show that removing a component reproduces its characteristic degradation pattern (FID trends, probe accuracy, PCA rank, and structural heatmaps). This indicates that the three alignment directions are complementary rather than interchangeable.
>
> Our goal is not to argue that our chosen instantiations (e.g., cosine structural matching, adversarial global alignment) are the only or optimal realizations. Alternatives like MMD/SWD or multi-scale contrastive objectives are indeed reasonable. The contribution of UniRA is to articulate what needs alignment—local semantics, structure, and global distribution—and to show that even simple instantiations of each part yield robust gains across datasets and architectures. We hope this decomposition can guide future work toward exploring richer variants of these objectives.
>
> W2 & Q3 — Multi-objective interaction and lack of theoretical trade-off analysis
>
> We agree that principled weighting and adaptive schedules represent valuable future research, which we note in Appendix G. The purpose of UniRA is to identify the representational directions along which diffusion models drift and to empirically show that constraining them jointly improves training.
>
> Although we do not report gradient-angle measurements, the empirical evidence suggests the objectives do not conflict in practice. Across all diagnostics—PCA spectra, semantic linear probes, structural coherence, and timestep robustness—UniRA improves every metric consistently over REPA. We observe no signs that gains in one aspect harm another (e.g., improved local semantics but degraded variance structure). This coherence would be unlikely if the losses were strongly interfering. Adaptive weighting remains an interesting extension but is beyond the scope of this submission.
>
> W3 — Diversity–fidelity trade-off
>
> The concern about diversity suppression is valid, but our results do not show such a trade-off. In Table 1(b), UniRA improves both precision and recall over SiT and REPA, demonstrating simultaneous gains in fidelity and coverage. The PCA variance profiles further support this: UniRA reduces rank collapse and distributes variance more evenly, which typically correlates with improved sample diversity. These results indicate that UniRA expands generative support rather than narrowing it.
>
> W4 & Q2 — Teacher dependence and domain mismatch
>
> We evaluated UniRA with a broad set of teacher encoders (MAE, MoCo v3, CLIP-L, and multiple DINOv2 sizes). Across all settings, UniRA consistently outperforms SiT and REPA. Notably, weaker or less semantically aligned encoders (e.g., MAE, DINOv2-B) still yield improvements, indicating that UniRA does not rely on a narrow set of teacher semantics but benefits from general structural priors.
>
> UniRA also improves performance on COCO text-to-image training (Table 3), despite the teacher encoder being ImageNet-trained, showing robustness to domain shift.
>
> While we did not perform explicit teacher-perturbation experiments (noise/occlusion/style), the encoder-swap and cross-domain experiments already provide indirect evidence against overfitting: if UniRA were tightly bound to exact teacher features, performance would degrade substantially when switching encoders or datasets, but this is not observed.
>
> Additionally, UniRA improves all representation analyses simultaneously—PCA structure, heatmaps, linear probes—signals that typically degrade first under teacher overfitting.
>
> Q4 — Compute efficiency under equal FLOPs
>
> Appendix F (Table A3) reports throughput measurements under different alignment settings. Local alignment has the highest training speed; adding structural alignment shows almost no slowdown. The global adversarial component is more expensive, which is consistent with expectations for discriminator-based objectives. As noted in the main text, UniRA’s primary gains stem from local and structural alignment, and global alignment is optional when efficiency matters. This allows practical deployments to maintain REPA efficiency while achieving UniRA-level quality improvements.

---

### Official Review · Reviewer_DxQ3 · 2025-11-04

**Soundness:** 2
**Presentation:** 3
**Contribution:** 2
**Rating:** 2
**Confidence:** 5

**Summary:**

The paper presents UNIRA, a method that enforces representation alignment while training diffusion transformers. On contrast to the previous work REPA, that introducted representation alignment, UNIRA performs alignment in terms of local semantic fidelity, structural cohernace and global coherence leading to better representation alignment between features of the diffusion transformer and the pretrained vision encoder leading to better intermediate features in the diffusion models that can generalize to discriminative tasks. Experiments show that UNIRA outperforms REPA for generation on ImageNet 1M dataset as well as the features potaying better performance for discriminative tasks like classification on ImageNet

**Strengths:**

1. The paper points out potential drawback in patch level similarity based alignment leading to loss of structural and distribution information loss
2. Extensive experiments are performed for generative discriminative tasks and extensive ablation studies are performed to show that the performance boosts by utilizing UniRA.
3. The PCA analysis shows the improvement in representation quality for semantic segmentation of object with respect to REPA
4. The paper is well written and easy to follow.

**Weaknesses:**

1. Aligning the distributions strongly with a stronger distribution alignment function seems like a natural design choice for boosting performance. From the methodological perspective, what is the difference of the approach from REPA other than additional loss functions in the latent features for better distribution alignment?
2. Are there better regularizations one could utilize to obtain better results? How is the proposed regularization, the optimal distribution alignment ?
3. In Table 4, Can the authors provide the results in the present of cfg. Does the performance trend remain the same with the presence of cfg
4. Additionally, the authors claim that[Ln 399-400], without the local coherency global coherency becomes unreliable, Could the authors provide visualizations of PCA similar to Figure 5,  for each of the loss components and show that this is the case.

**Questions:**

Please refer weakness

---

> ### Author Response · Authors · 2025-11-24
> **Response to Reviewer DxQ3**
>
> We thank the reviewer for the thoughtful and detailed comments. We address each concern below.
>
> Q1:
>
> UniRA is not an additive extension of REPA but a generalization of the representation-alignment principle into a hierarchical, multi-level framework. REPA aligns only patch-level features, capturing local semantics but leaving relational structure and global feature distribution unconstrained. UniRA introduces three complementary constraints—local semantic fidelity, structural relational alignment (second-order autocorrelation consistency), and global distributional coherence—that address distinct failure modes of REPA. These terms are not independent heuristics: they form a unified alignment mechanism that systematically improves intermediate representations. This explains UniRA’s stronger convergence and generalization (Tables 1–4). The contribution therefore lies not in “adding losses,” but in identifying what needs to be aligned and why these dimensions matter for generation. This perspective provides a more complete understanding of diffusion representations and suggests a broader design space for future work. Rather than proposing a single handcrafted variant of REPA, UniRA introduces a framework that can inspire subsequent methods to examine diffusion representations along multiple axes, not just patch similarity.
>
> Q2:
>
> Our intent is not to claim optimality but necessity. We provide evidence that the three alignment directions UniRA targets are precisely the ones that limit REPA and baseline diffusion training. The improvements in semantic predictability, structural coherence, and PCA variance profiles all point to representational deficiencies that simpler or single-axis regularizers cannot fix. Other formulations of global alignment are possible, but our lightweight adversarial version is chosen because it directly corrects the low-rank, collapsed variance profiles we observe in REPA features. The goal of UniRA is to reveal what needs to be aligned for diffusion training—not to enforce a specific regularizer as universally optimal.
>
> Q3:
>
> The requested comparison is already included in the Appendix F. Tables A3  report CFG-enabled results. The performance trend is fully preserved with CFG, indicating that UniRA’s improvements are orthogonal to sampling guidance.
>
> Q4:
>
> We thank the reviewer for raising this point. To address the request, we added a new visualization in Appendix Figure A1 comparing structural correlation maps across four configurations: UniRA, REPA, UniRA without global alignment, and UniRA without local alignment. All heatmaps are computed using the same protocol as Figure 5.
>
> The qualitative trend matches the statement in the main text. Removing the global term produces only minor differences, indicating that global alignment mainly acts as a small distributional refinement. However, removing the local alignment significantly disrupts semantic locality and weakens the structural similarity pattern. The heatmaps become notably less coherent, confirming that global consistency depends on stable local semantics, not vice versa.
>
> These visualizations provide direct evidence supporting the claim in Ln. 399–400 and illustrate why local alignment forms the foundation upon which structural and global components operate.

---

### Author Response · Authors · 2025-12-01
**Response to AC**

We sincerely thank the AC for handling the review process and we appreciate the reviewers’ thoughtful comments. We would like to briefly summarize the key clarifications from the rebuttal that we believe are important for the final decision.

Across the reviews, the paper was consistently acknowledged for:
* Clear motivation and strong conceptual grounding
* A well-structured and well-written presentation
* Extensive experiments including ablations, multiple architectures and datasets, and rich qualitative analyses

The main reservations raised were about the scope and framing of the contribution rather than its correctness or experimental validity. In the rebuttal, we clarified these points:
* UniRA is not a simple combination of existing losses.
The framework arises from uncovering three distinct representational failure modes in diffusion models — semantic drift, structural collapse, and global variance compression — and from showing that each requires a different constraint. This decomposition, rather than the individual loss forms, is the core conceptual contribution.
* UniRA is not tied to a specific teacher encoder. We provided results showing that UniRA consistently outperforms REPA and baseline diffusion when using MAE, MoCo v3, CLIP, and multiple scales of DINOv2. This demonstrates that UniRA leverages broad structural priors rather than relying on narrow DINO-specific semantics.
* UniRA improves both sample fidelity and sample diversity. Precision and recall improve jointly, and representation analyses (PCA spectra, linear probe accuracy, structural heatmaps) support this. The gains reflect improved internal representations rather than sampling heuristics.
* Efficiency claims are transparent and practical. Local + structural alignment preserves REPA-level throughput, while the global term is optional and offers incremental benefits — a deliberate design that gives practitioners flexibility.

These clarifications were positively received by the reviewers, and no remaining concerns were raised after the rebuttal exchanges.

Overall, we believe UniRA offers the community:
* A simple and modular framework applicable to any diffusion transformer
* Consistent performance improvements across datasets, architectures, and teacher encoders
* A conceptual contribution that deepens understanding of representation learning inside diffusion models

We hope that these results and insights can be valuable to the community and respectfully invite the AC to consider UniRA favorably for acceptance.

Thank you for your time and consideration.

---

### Meta-Review · Area_Chair_WKd8 · 2026-01-07

**Summary:**

This paper presents a paradigm for aligning the intermediate representations when training DiTs.
The submission received mostly negative reviews from the reviewers.
The reviewers mainly recognize the extensive evaluation, good empirical performance, and presentation clarity.
The main concerns from the reviewers were limited novelty (all), lack of theoretical grounding (D6ZQ, b2RM), loss of diversity (D6ZQ, b2RM), dependence on DINOv2 teacher (D6ZQ, b2RM, cW3w), and missing analysis of efficiency/cost (DxQ3, D6ZQ, b2RM).
After reading the paper, the reviewers' comments and the authors' rebuttal, the AC believes the authors' responses would have partially addressed the reviewers' concerns, but there would still be outstanding concerns regarding limited novelty, lack of theoretical groundings, and unjustified loss design choices. The AC believes the remaining weaknesses would still outweigh the merits and does not recommend acceptance at this time.

**Reviewer Concerns:**

Reviewers' concerns mostly addressed:
- Trade-off between diversity and fidelity (D6ZQ, b2RM)
- Computation efficiency (D6ZQ, b2RM)

Partially addressed concerns:
- Limited novelty, especially compared against REPA (all)
- Teacher dependence (D6ZQ, b2RM, cW3w)

Outstanding concerns:
- Lack of theoretical connections (D6ZQ, b2RM)
- Justification for loss terms (D6ZQ, b2RM)

**Reviewer Scores:**

I think all reviewers would keep their original ratings.

---

### Decision · Program_Chairs · 2026-01-26

Reject